# Neurotransmitter identity is acquired in a lineage-restricted manner in the *Drosophila* CNS

Haluk Lacin[1,2]*, Hui-Min Chen[1], Xi Long[1], Robert H Singer[1,3], Tzumin Lee[1], James W Truman[1,4]

[1]Janelia Research Campus, Howard Hughes Medical Institute, Ashburn, United States; [2]Department of Genetics, Washington University, Saint Louis, United States; [3]Department of Anatomy and Structural Biology, Albert Einstein College of Medicine, Bronx, United States; [4]Friday Harbor Laboratories, University of Washington, Friday Harbor, United States

**Abstract** The vast majority of the adult fly ventral nerve cord is composed of 34 hemilineages, which are clusters of lineally related neurons. Neurons in these hemilineages use one of the three fast-acting neurotransmitters (acetylcholine, GABA, or glutamate) for communication. We generated a comprehensive neurotransmitter usage map for the entire ventral nerve cord. We did not find any cases of neurons using more than one neurotransmitter, but found that the acetylcholine specific gene ChAT is transcribed in many glutamatergic and GABAergic neurons, but these transcripts typically do not leave the nucleus and are not translated. Importantly, our work uncovered a simple rule: All neurons within a hemilineage use the same neurotransmitter. Thus, neurotransmitter identity is acquired at the stem cell level. Our detailed transmitter- usage/lineage identity map will be a great resource for studying the developmental basis of behavior and deciphering how neuronal circuits function to regulate behavior.
DOI: https://doi.org/10.7554/eLife.43701.001

*For correspondence:
lacinhaluk@gmail.com

## Introduction

The ventral nerve cord (VNC) of *Drosophila melanogaster* is home to circuits coding for vital behaviors, such as walking, jumping, and flight (*Harris et al., 2015*; *Gowda et al., 2018*). It is composed of about 16,000 neurons, all of which arise from a set of segmentally repeated 30 paired and one unpaired neural stem cells (Neuroblasts [NBs]) (*Bossing et al., 1996*; *Schmid et al., 1999*; *Lacin and Truman, 2016*; *Shepherd et al., 2016*). NBs generate unique progeny via undergoing two rounds of proliferation: a brief embryonic and an extended postembryonic phase (*Truman and Bate, 1988*; *Truman et al., 2004*; *Truman et al., 2010*). Embryonic neurogenesis generates the neurons of the larval CNS and many of these cells are then remodeled to function in the adult CNS (*Truman, 1990*; *Hartenstein and Wodarz, 2013*). 90–95% of the adult neurons, however, are adult-specific and arise during the post-embryonic phase of neurogenesis.

The VNC contains only NBs that show a Type I pattern of proliferation (*Truman et al., 2010*). Each NB divides repeatedly via asymmetric cell division to renew itself and to generate a secondary precursor cell, called a ganglion mother cell (GMC) (*Figure 1A*). Each GMC, in turn, divides terminally to form two neurons, each of which acquires a unique identity due to the presence or absence of active Notch signaling (*Spana and Doe, 1996*; *Skeath and Doe, 1998*; *Truman et al., 2010*). Within a NB progeny, the Notch-ON neurons are called the 'A' hemilineage; their Notch-OFF siblings are called the 'B' hemilineage (*Figure 1B*).

**Figure 1.** Strategies used to delineate neurotransmitter identity. (**A**) A schematic representation of the NB division mode. With each division, the NB renews itself and generates a GMC, which divides via asymmetric cell division to generate two neurons. One of these neurons receives Notch (**N**) signaling and adopts 'A' fate (green), and the other neuron does not receive Notch signaling and adopts the 'B' fate (magenta). (**B**) Repeated NB divisions generate two hemilineages: The Notch ON 'A' hemilineage and the Notch OFF 'B' hemilineage. Neurons born in late embryonic and postembryonic stages adopt similar fates within a hemilineage whereas early embryonic divisions generate diverse neuronal types (asterisks). (**C**) A schematic representation of the adult VNC and hemilineages coming from NB1-2 (1A and 1B) and NB7-1 (3A and 3B). Neurons that belong to the same

*Figure 1 continued on next page*

*Figure 1 continued*

postembyonic hemilineage cluster together and send their cellular processes to the same region, while the neurons of the sibling hemilineage extend their axons to a different region (compare 1A to1B or 3A to 3B). (D) RNA in situ hybridization against neurotransmitter specific genes *ChAT* (green; cholinergic), *gad1*(blue; GABAergic), and *VGlut* (magenta; glutamatergic) reveals mutually exclusive mRNA expression patterns. Note that neurons with the same neurotransmitter cluster together. (E) Hb9 expression (green) marks neurons of three hemilineages (4B, 10B, and 16B) in the adult VNC. (F–J) Split-GAL4 intersectional strategy with genetic tools reporting the expression of lineage-marking transcription factors and neurotransmitter specific genes. *Nkx6*$^{DBD}$-*VGlut*$^{AD}$ marks 15B neurons (F), *ey*$^{AD}$-*VGlut*$^{DBD}$ marks 8A and 21A neurons (G); *msh*$^{AD}$-*ChAT*$^{DBD}$ marks 1A and 14A neurons (H); *dbx*$^{DBD}$-*gad1*$^{AD}$ marks 13A and 19A neurons (I); and *msh*$^{AD}$-*VGlut*$^{DBD}$ marks 14A and 21A neurons (J). (K) Antibody labeling against ChAT (magenta) and GABA (blue) does not mark glutamatergic 15B leg motor neurons (green), which were visualized via NB intersected reporter immortalization (IM) of the R10C12 GAL4 driver. Solid lines indicate midline. Scale bar is 10 microns.

DOI: https://doi.org/10.7554/eLife.43701.002

The following figure supplement is available for figure 1:

**Figure supplement 1.** *msh*$^{AD}$-*ey*$^{DBD}$ split GAL4 combination marks NB4-3 and its progeny (white arrows) in the VNC (A–D), and a local antennal lobe lineage, ALv2 (magenta arrowheads) in the brain (A, E, F) across all life stages.

DOI: https://doi.org/10.7554/eLife.43701.003

By progressing through the temporal transcriptional cascade, Hunchback →Kruppel → Pdm, NBs generate diverse 'A' and 'B' neurons during the early embryonic phase (*Figure 1B*), (*Kambadur et al., 1998*; *Isshiki et al., 2001*). Subsequently, NBs express Castor and Grainyhead in late embryonic stages and many NBs maintain this Castor/Grainyhead expression into the postembryonic stages (*Almeida and Bray, 2005*; *Maurange et al., 2008*; *Lacin and Truman, 2016*). Correlated with this shared gene expression, neurons of a hemilineage ('A' or B') that are born during the late embryonic and early postembyonic stages often adopt similar fates (*Truman et al., 2010*; *Lacin and Truman, 2016*). Recent studies characterized the morphologies of postembryonic hemilineages in their immature states in the larva and mature states in the adult (*Truman et al., 2004*; *Truman et al., 2010*; *Harris et al., 2015*; *Shepherd et al., 2016*). These studies revealed that in the larva, the immature neurons of each hemilineage cluster together and extend their initial processes as a bundle to the same region and that after metamorphosis, in the adult, they continue to be clustered and share common anatomical and functional features (*Figure 1C*).

In addition to similar morphology, neurons within a postembryonic hemilineage share patterns of transcription factor expression. In the larval VNC, each hemilineage cluster can be identified with a specific combination of transcription factor expression (*Lacin et al., 2014*). Interestingly, vertebrate homologs of many of these hemilineage-specific transcription factors are expressed in the spinal cord and are required for fate determination (*Thor and Thomas, 1997*; *Jessell, 2000*). For example, in flies, the combinatorial expression of Lim3, Islet, and Nkx6 is observed uniquely in hemilineage 15B, which is composed of leg motor neurons (*Lacin et al., 2014*). In mice, homologs of these three factors are essential for the identity of spinal motor neurons (*Tsuchida et al., 1994*; *Lacin et al., 2014*). Indeed, interneurons in the vertebrate spinal cord are also organized into discrete cardinal classes that share developmental origins, transcription factor and neurotransmitter expression, and functional roles (*Briscoe et al., 2000*; *Jessell, 2000*; *Zhang et al., 2008*; *Arber, 2012*; *Lu et al., 2015*). The fly VNC appears to be organized in an analogous manner, with the developmental origins of neuronal clusters providing the basis for their transcription factor expression and functional properties. The relationship between specific stem cells and neurotransmitter expression in their progeny, though, has yet to be resolved.

Studies on grasshoppers (*Siegler et al., 2001*) and *Manduca sexta* (*Witten and Truman, 1991*) showed that clusters of GABAergic interneurons were based on their lineage of origin. Likewise, cholinergic and glutamatergic neurons are also typically found as clusters in the VNC and the brain consistent with a shared lineage (*Salvaterra and Kitamoto, 2001*; *Liu and Wilson, 2013*) In the fly, as a prelude to studies that seek to dissect the developmental basis of behavior, we wanted to map neurotransmitter usage comprehensively across the VNC to determine how neurotransmitter selection relates to lineage identity. A similar comprehensive neurotransmitter map was generated for the *C. elegans* nervous system and proved to be beneficial in identifying regulatory mechanisms that control neurotransmitter identity and circuit assembly (*Serrano-Saiz et al., 2013*; *Pereira et al., 2015*; *Gendrel et al., 2016*; *Serrano-Saiz et al., 2017*). Here, by using molecular and genetic tools, we mapped neurotransmitter usage in all hemilineages in the adult fly VNC. Our results revealed that,

as found in the neurons of the vertebrate cardinal classes, all neurons within a fly hemilineage use the same neurotransmitter. In agreement with our earlier findings (*Harris et al., 2015*), this study further shows that hemilineages are not just developmental units but also functional units that drive animal behavior.

## Results

The vast majority of fly neurons use one of the three fast acting neurotransmitters: GABA, glutamate, or acetylcholine. First, we wanted to know the relative usage of these neurotransmitters in the VNC and asked if neurons use more than one neurotransmitter, a possibility suggested for the fly brain based on transcriptomic studies (*Croset et al., 2018*). We visualized mRNAs of neurotransmitter-specific genes *glutamate decarboxylase 1* (*gad1*), *vesicular glutamate transporter* (*VGlut*), and *choline acetyltransferase* (*ChAT*) in the same animal via RNA in situ hybridization (*Long et al., 2017*). *gad1* and *ChAT* encode enzymes that are required for the production of GABA and acetylcholine, respectively, while *VGlut* encodes a glutamate transporter (*Slemmon et al., 1991*; *Kulkarni et al., 1994*; *Daniels et al., 2004*). We observed mutually exclusive expression patterns among these mRNAs across the entire VNC (*Figure 1D*), suggesting that most if not all neurons in the VNC use only one of these neurotransmitters. One caution, though, is that some of the GABAergic and glutamatergic neurons show low levels of ChAT transcripts. We address this issue below.

Considering the obvious clustering of neurons expressing a given transmitter type in the VNC (*Figure 1D*), we asked if these clusters correspond to the postembryonic hemilineages and if all the neurons within a hemilineage use the same neurotransmitter. Thus, we decided to characterize all postembryonic hemilineages in terms of their choice of neurotransmitters.

### Visualizing adult lineages

We employed several different approaches to visualize specific hemilineages in the adult VNC, each of which can be identified based on cell body position and axonal projection (*Harris, 2013*; *Harris et al., 2015*; *Shepherd et al., 2016*). First, we used a set of transcription factors that are lineage-specific molecular markers that identify immature neurons of specific hemilineages in the larval VNC (*Lacin et al., 2014*). Expression of many of these factors are maintained through metamorphosis and identify the corresponding mature neurons in the adult (*Figure 1E*; *Figure 2*). We also generated split-GAL4 halves reporting the expression of some of these transcription factors via converting available MIMIC lines with the TROJAN method or via CRISPR tagging (*Venken et al., 2011*; *Diao et al., 2015*). We used these split-GAL4 lines to reveal overlapping expression patterns. Simply, we generated reporter lines expressing either the DNA-binding domain (DBD) or activation domain (AD) of the GAL4 transcription factor with the idea that both the AD and DBD need to be present in the same cell to form a functional GAL4 reporter.

We used these genetic lines in combination with one another or with neurotransmitter-specific split-GAL4 halves (*Diao et al., 2015*) to mark specific hemilineages and reveal their neurotransmitter identity (*Figure 1F–J*). Of note, intersectional combinations of these split -GAL4 halves proved to be extremely useful for developmental studies because they mark specific hemilineages throughout the development and adult life unlike many of the enhancer driven GAL4 lines from the Janelia (*Jenett et al., 2012*) or Vienna (*Kvon et al., 2014*) collection, which show dynamic expression through metamorphosis (*Figure 1—figure supplement 1*). In addition to lineage specific transcription factor expression, we employed NB intersected reporter immortalization to visualize specific neuronal lineages in the adult (*Figure 1K*: *Awasaki et al., 2014*; *Lacin and Truman, 2016*). This technique employs a cascade of recombinases and allows us to visualize the entire progeny of a NB when used with NB-specific GAL4 lines (*Table 1*). Simply, it maintains (i.e. immortalizes) transient reporter expression in the NB and forces the progeny of the NB to express the reporter in an irreversible and lineage-specific manner. This technique can unambiguously identify hemilineages in which the sibling hemilineage was eliminated by apoptosis (e.g. 2A, 10B, and 17A) without any additional markers since they are the only surviving lineages (*Figure 2*; *Truman et al., 2010*). For the lineages in which both hemilineages survive, we used molecular markers, the location of cell bodies, and axonal projections to distinguish between sibling hemilineages (*Lacin et al., 2014*; *Harris et al., 2015*; *Shepherd et al., 2016*).

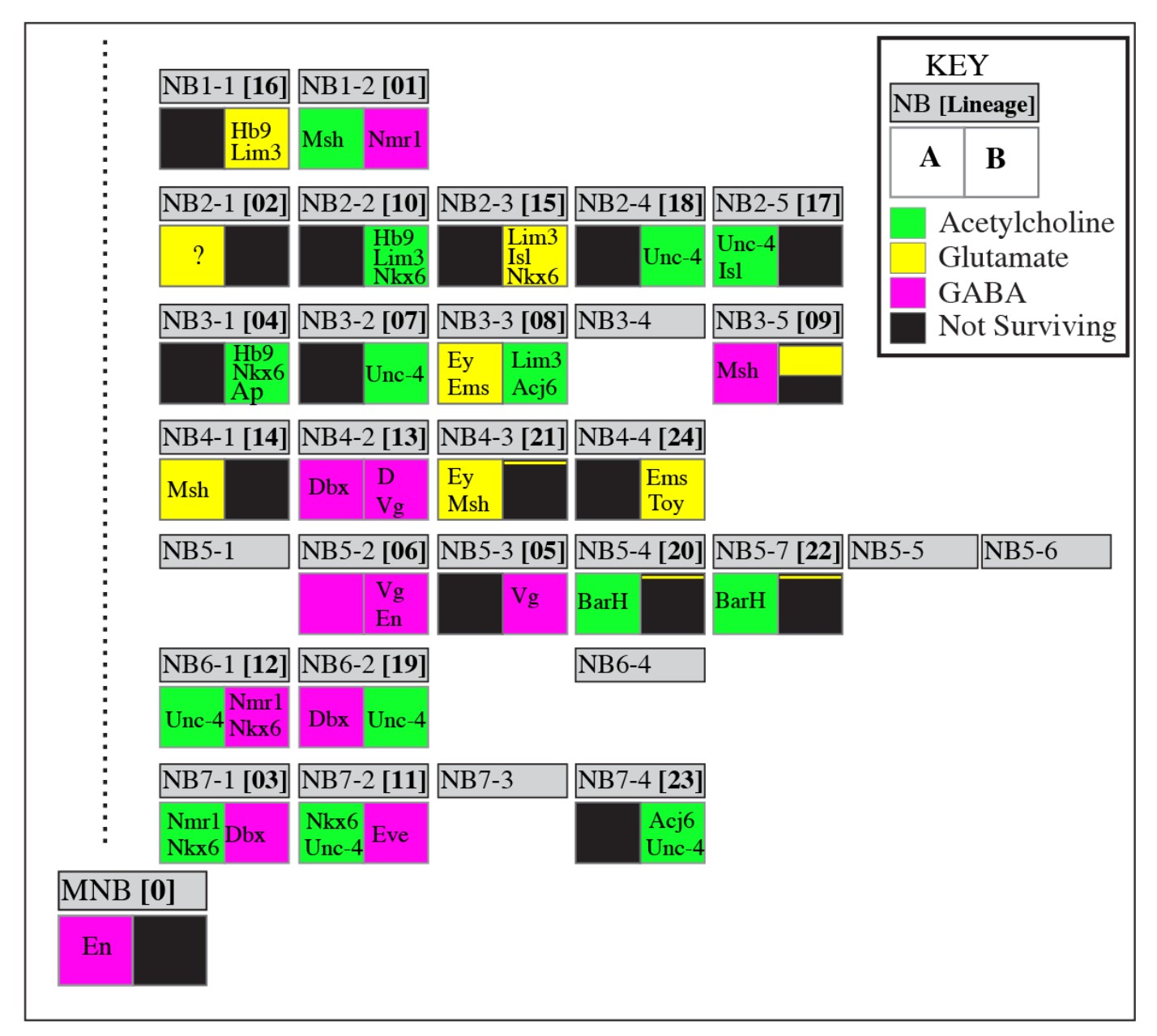

**Figure 2.** Neurotransmitter map of the adult *Drosophila* VNC. NB identity and corresponding postembryonic lineages are shown in gray boxes. The left bottom box represents the 'A' hemilineage and the right bottom box represents the 'B' hemilineage. Out of 34 major hemilineages that survive to function in the adult nervous system, 14 are cholinergic (green), eight are glutamatergic (yellow) and 12 are GABAergic (magenta). Many hemilineages are eliminated completely by programmed cell death (black boxes) except in a few cases (9B, 21B, and 20/22B) where a small number of neurons survive. Hemilineage specific transcription factor expressions are stated in each box. These transcription factors are expressed in most if not all neurons of the indicated hemilineages. Note that Bar H expression (asterisks) is based on studies in larval stages; its adult expression was not tested due to lack of reagents. Dbx expression in 3B neurons (double asterisks) occurs transiently in larval stages, but does not last into the adult. NBs 5–1, 5–5, 5–6, 6–4, and 7–3 are eliminated by apoptosis during late embryonic development so they do not have postembryonic progeny. NB3-4 generates a few postembryonic motor neurons; their incorporation to the adult VNC is unknown. Note that the correspondence of postembryonic lineages to the NB identity is based on *Lacin and Truman (2016)*, and differs from the map of *Birkholz et al. (2015)*.

DOI: https://doi.org/10.7554/eLife.43701.004

**Table 1.** GAL4 lines used for NB intersected reporter immortalization.

| Line | Neuroblast | Lineages |
| --- | --- | --- |
| 16A05AD;28H10DBD | NB1-2 | 1A-1B |
| 70D06AD;28H10DBD | NB2-1 | 2A |
| 10C12 | NB2-3 | 15B |
| 65G02 | NB2-4 | 18B |
| 19B03 | NB2-5, NB2-4 | 17A, 18B |
| 21E09AD;16H11DBD | NB3-2 | 7B |
| ems-GAL4 | NB2-2, NB3-3, NB3-5 | 10B, 8A-8B, 9A-9B |
| 59E09 | NB3-5 | 9A-9B |
| VT48571 | NB4-1 | 14A |
| 81C12A;42F01DBD | NB4-2 | 13A-13B |
| 16H11AD-19B03DBD | NB4-3 | 21A |
| VT205490 | NB4-4 | 24A |
| 45D04 | NB5-2 | 6A-6B |
| 54B10 | NB5-3 | 5B |
| 19B03AD;45D04DBD | NB5-4, NB5-7 | 20-22A |
| 81F01 or 70D06AD;42F01DBD | NB6-1 | 12A-12B |
| R76D11 | NB6-2 | 19A-19B |
| R51B04 | NB7-1 | 3A-3B |
| 35B12AD;28H10DBD | NB7-2 | 11A-11B |
| 19B03AD;18F07DBD | NB7-4 | 23B |
| 70D06 | MNB | 0B |

For detailed information about these lines, see *Lacin and Truman, 2016*.

DOI: https://doi.org/10.7554/eLife.43701.005

Utilizing the tools mentioned above we systematically visualized each hemilineage in the VNC and then mapped the fast-acting neurotransmitter used by the neurons of each hemilineage (*Figure 2*). We found that all neurons within a hemilineage use the same neurotransmitter (see below). Of the 34 major hemilineages per thoracic ganglia, eight hemilineages are Glutamatergic, 12 hemilineages are GABAergic, and 14 hemilineages are cholinergic. For easier access for the readers, the data for each hemilineage will be presented below in groups based on the neurotransmitter type although the neurotransmitter identity of each hemilineage was identified independently in an unbiased fashion.

## Glutamatergic lineages

Glutamate can serve as an inhibitory or excitatory neurotransmitter in the fly CNS depending on the neuronal type (*Daniels et al., 2004*; *Liu and Wilson, 2013*). To identify neurons and neuronal lineages that use glutamate, we used genetic reporters- *VGlut*-GAL4, VGlut[DBD], *VGlut*-LexA (*Diao et al., 2015*) and *VGlut*[AD] (this study), and a vGLUT specific antibody (*Daniels et al., 2004*). *Figure 3* shows an example of how we used multiple approaches to identify the neurotransmitter identity of neurons in the 14A and 21A hemilineages. Msh marks lineages 1A, 9A, 14A and 21A in both the larval and adult nerve cords (*Lacin and Truman, 2016*; this study). We used *msh*[AD] in combination with VGlut[DBD] to see if any of the Msh+ lineages were glutamatergic as GAL4 mediated expression will only occur in cells that co-express both transgenes. This split-GAL4 combination identified the 14A (67 ± 2 neurons; n = 5 hemisegments) and 21A (63 ± 1 neurons; n = 3 hemisegments) hemilineages as likely glutamatergic (*Figure 3A,B*). The VGlut-LexA line similarly marked the 14A and 21A neurons, which were revealed by Msh antibody staining (*Figure 3C* and not shown). We confirmed these findings with an antibody staining against VGlut and showed that all 14A and 21A neurons do indeed express VGlut protein (*Figure 3D*; *Figure 4—figure supplement 1D,G*).

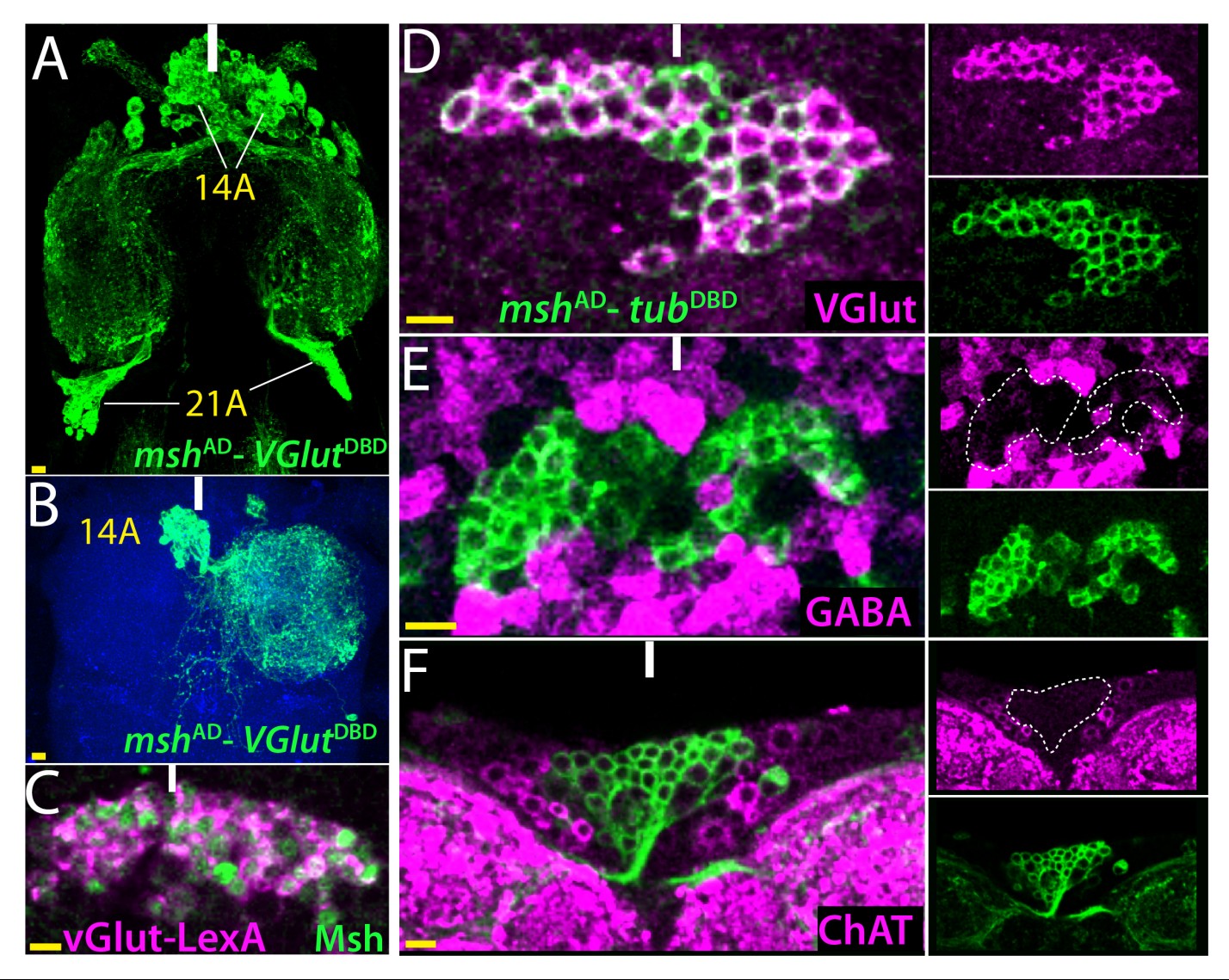

**Figure 3.** 14B neurons are glutamatergic. (A) The split GAL4 combination of *msh*[AD]-*VGlut*[DBD] marks 14A and 21A neurons. (B) A lineage flip-out clone generated with *msh*[AD]-*VGlut*[DBD] visualizes 14A neurons, which innervate the contralateral leg neuropil. (C) *VGlut*-LexA driver labels 14A neurons, which are revealed by the Msh expression. (D–F) *msh*[AD]-*tub*[DBD] is used to visualize bilaterally paired 14A clusters; single confocal stack is shown. VGlut antibody labels 14A neurons (D) but GABA and ChAT antibodies fail to mark them (E, F). Images in (A, B) are maximum projections while the rest are from a single confocal stack. Solid lines indicate midline. Scale bar is 10 microns.

DOI: https://doi.org/10.7554/eLife.43701.006

Interestingly, a *msh*[AD]-*ChAT*[DBD] combination also marked 14A neurons (*Figure 1H*), suggesting 14A neurons might also be cholinergic. To resolve such discrepancies, we stained the nerve cords against ChAT and GABA and failed to detect any cholinergic or GABAergic neurons among the 14A or 21A neuronal populations (*Figure 3E,F*; *Figure 4—figure supplement 1G*). Thus, our results clearly show that all neurons within 14A and 21A hemilineages acquire a glutamatergic fate. The interesting mismatch between ChAT reporters and the immunocytochemical staining will be addressed below.

Motor neurons in *Drosophila* use glutamate as the choice of neurotransmitter for muscle excitation (*Johansen et al., 1989*). 15B and 24A hemilineages are made of entirely motor neurons, which innervate the leg musculature (*Truman et al., 2004*; *Baek and Mann, 2009*; *Truman et al., 2010*). As expected, antibodies detected vGLUT protein in the motor neurons of lineage 15B and 24A,

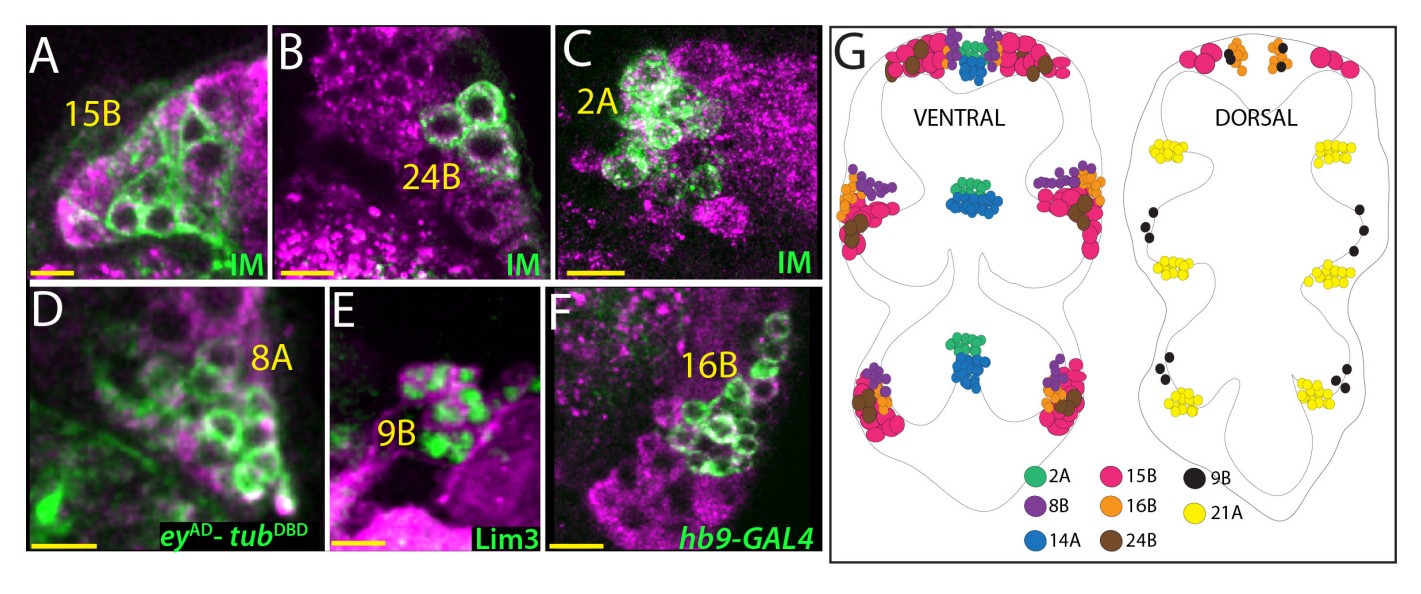

**Figure 4.** Glutamatergic lineages. VGlut antibody (magenta in **A, B, C, D, and F**) and VGlut-LexA reporter (magenta in **E**) label neurons of 15B, 24B, 2A, 8A, 9B, and 16B hemilineages in the adult VNC. NB intersected reporter immortalization (IM) with the R10C12, VT205490, and R70D06$^{AD}$-R28H10$^{DBD}$ driver lines used to visualize 15B, 24B and 2A neurons, respectively (**A, B, and C**). $ey^{AD}$-$tub^{DBD}$, Lim3, and $hb9$-GAL4 expressions used to mark 8A, 9B, and 16B neurons, respectively (**D, E, and F**). (**G**) The location of glutamatergic hemilineages shown schematically via color-coded map. Left and right images represent ventral and dorsal halves of the VNC, respectively. Only thoracic lineages are shown. All images are from a single confocal stack. See *Figure 4—figure supplement 1* for individual channels. Scale bar is 10 microns.

DOI: https://doi.org/10.7554/eLife.43701.007

The following figure supplement is available for figure 4:

**Figure supplement 1.** Glutamatergic hemilineages.

DOI: https://doi.org/10.7554/eLife.43701.008

which were visualized via NB intersected reporter immortalization (*Figure 4A,B*; *Table 1*), but neither GABA nor ChAT protein was detected in these lineages (*Figure 1—figure supplement 1E,H*). Transcription factor expression profile for adult 24B neurons was studied in detail, and combinatorial expression of six transcription factors was shown to be responsible for the identity of the 24B motor neurons (*Enriquez et al., 2015*). 15B motor neurons express Lim3, Nkx6, and Islet (Isl) during larval development (*Lacin et al., 2014*). Based on antibody staining and genetic reporters (*lim3*$^{DBD}$ and *nkx6*$^{DBD}$), we found that most of the mature 15B motor neurons in the adult co-express Lim3, Isl and Nkx6 (*Figure 1F*; *Figure 4—figure supplement 1E*). Interestingly, the level of Nkx6 expression was not uniform across 15B neurons: some neurons expressed it at high levels while some expressed it at low level (*Figure 4—figure supplement 1E*). The amount of Nkx6 might be regulating differentiation of 15B neurons.

In addition to lineages 14A, 15B, 21A and 24A, we identified four more glutamatergic hemilineages: 2A, 8A, 9B, and 16B (*Figure 4G*). NB2-2 generates only 2A neurons since their 2B sibs die soon after birth (*Truman et al., 2010*). We used NB2-2 intersected reporter immortalization (*Table 1*) or R50G08-GAL4 expression to mark 2A neurons, which reside medially and in a close proximity to 14A neurons. Labeling with both antibodies and genetic tools showed that all 2A neurons (34 ± 3, n = 3 hemisegments) express vGLUT, but neither ChAT nor GABA (*Figure 4C*; *Figure 4—figure supplement 1A*).

Neurons of the 8A cluster are located laterally and innervate the leg neuropil (*Harris et al., 2015*). We visualized the 8A neurons by $ey^{DBD}$-$tub^{AD}$ or OK107-GAL4, a reporter line inserted into *ey* gene (*Adachi et al., 2003*). We observed 55 ± 2 (n = 5 hemisegments) 8A neurons per cluster and all appeared to express VGlut (*Figure 4D*; *Figure 4—figure supplement 1B*). Moreover, the intersection of $ey^{AD}$-$VGlut^{DBD}$ drivers also marked the 8A neurons further confirming their

glutamatergic identity (*Figure 1G*). In agreement with these results, we did not detect any ChAT or GABA expression in neurons of the 8A hemilineage (*Figure 4—figure supplement 1B*).

It was previously thought that only the 'A' sibs in lineage nine survived and that this 9A cluster included a large group of ipsilateral cells and a much smaller group of cells that projected to the contralateral side of the segment (*Truman et al., 2010*). We found, though, that the contralateral cells are actually surviving neurons of the 9B hemilineage based on wild type lineage clones and MARCM clones of Notch signaling mutants (*Figure 4—figure supplement 1C*; not shown). 9B hemilineage includes 18 ± 3 neurons (n = 4 hemisegments) that express Lim3 and Acj6 in the adult VNC (*Figure 4E* and not shown). Both VGlut antibody and VGlut-LexA mark 9B neurons (*Figure 4—figure supplement 1C*).

Neurons of the 16B hemilineage express Hb9 and Lim3 in the adult VNC and are located laterally to the other cluster of Hb9+ Lim3+ neurons that constitute the 10B cluster (*Figure 4—figure supplement 1F*; *Shepherd et al., 2016*). Both VGlut antibody and VGlut-LexA driver mark the 16B neurons (44 ± 4, n = 3 hemisegments), and ChAT and GABA antibodies fail to mark them (*Figure 4F*; *Figure 4—figure supplement 1A*).

In conclusion, we identified eight hemilineages that use glutamate as their neurotransmitter (*Figure 4G*). Two clusters are composed of leg motor neurons, and the other six are composed of interneurons. Five of these interneuronal hemilineages (8A, 9B, 14A, 16B, and 21A) innervate the leg neuropil while the 2A neurons innervate flight-related neuropils.

## GABAergic lineages

GABA is the major fast inhibitory neurotransmitter in the *Drosophila* CNS (*Wilson and Laurent, 2005*). To identify GABAergic neurons, we used genetic reporters (*gad1*[AD], *gad1*-LexA (*Diao et al., 2015*); *gad1*[DBD](this study)) and GABA specific antibodies. In total, we identified 12 GABAergic lineages: 0A, 1B, 3B, 5B, 6A, 6B, 9A, 11B, 12B, 13A, 13B, and 19A (*Figure 5K*).

The local interneurons produced by the median NB (MNB) are reported to be GABAergic throughout insects (*Goodman and Spitzer, 1979*; *Goodman et al., 1980*; *O'Dell and Watkins, 1988*; *Thompson and Siegler, 1991*; *Witten and Truman, 1991*; *Thompson and Siegler, 1993*; *Bossing and Technau, 1994*; *Witten and Truman, 1998*; *Schmid et al., 1999*; *Shepherd et al., 2016*). The 0A neurons, which derive from the MNB, in *Drosophila* were cleanly labeled by NB intersected reporter immortalization and Engrailed/Invected expression (*Table 1*; *Figure 5—figure supplement 1A*). We detected 50 ± 3 (n = 4) 0A neurons in the T1 and T2 hemisegments and 30 neurons (n = 1) in the T3 hemisegment. These neurons were immunopositive for GABA and immunonegative for ChAT (*Figure 5A*; *Figure 5—figure supplement 1A*). Consequently, as in other insects, postembryonic progeny of the MNB in *Drosophila* are also GABAergic.

The 1B neurons are generated in the anterior border of the T2, T3, and A1 segments (*Truman et al., 2004*), but their cell bodies are pulled anteriorly during metamorphosis and become situated on the posterior border of the next anterior segment in the adult (*Harris et al., 2015*). 1B neurons are local interneurons that innervate the leg neuropil. We immortalized the reporter expression in the progeny of NB1-2 and visualized the 1A and 1B neurons (*Table 1*). The 1B hemilineage can easily be distinguished from the 1A hemilineage in the adult due to their segmental separation. The 1B cluster contained 56 ± 2 neurons (n = 5 hemisegments), and 37 ± 2 (n = 5) of them express Nmr1 (*Figure 5—figure supplement 1B*). We found that GABA, but never ChAT, immunostaining marks 1B neurons (*Figure 5B*; *Figure 5—figure supplement 1B*). Thus, our results show that the 1B neurons are GABAergic.

We visualized the entire progeny of NB7-1, which are the 3A and 3B neurons, via NB intersected reporter immortalization (*Table 1*). The 3A neurons express Nkx6 during both larval and adult stages (*Lacin et al., 2014*). Many of the 3B neurons express Dbx during larval development, however after pupa formation their Dbx expression disappears (*Lacin et al., 2014*). Within the NB7-1 lineage we assumed that Nkx6 expressing cells are the 3A neurons and that Nkx6 negative cells are the 3B neurons. We found that the Nkx6-negative 3B neurons express GABA, while the Nkx6-positive 3A neurons express ChAT (*Figure 5—figure supplement 1C*; *Figure 6B*). We detected 41 ± 3 (n = 3) 3B neurons per T1 cluster. We expect a similar number of 3B neurons for T2 segments, but did not obtain quantifiable clones. T3 segments did not appear to have any 3B neurons as previously suggested (*Harris et al., 2015*). To confirm that the 3B neurons are GABAergic, we stained larval nerve cord expressing gad1-LexA reporter with the Dbx antibody. As expected, we found that *gad1*-LexA

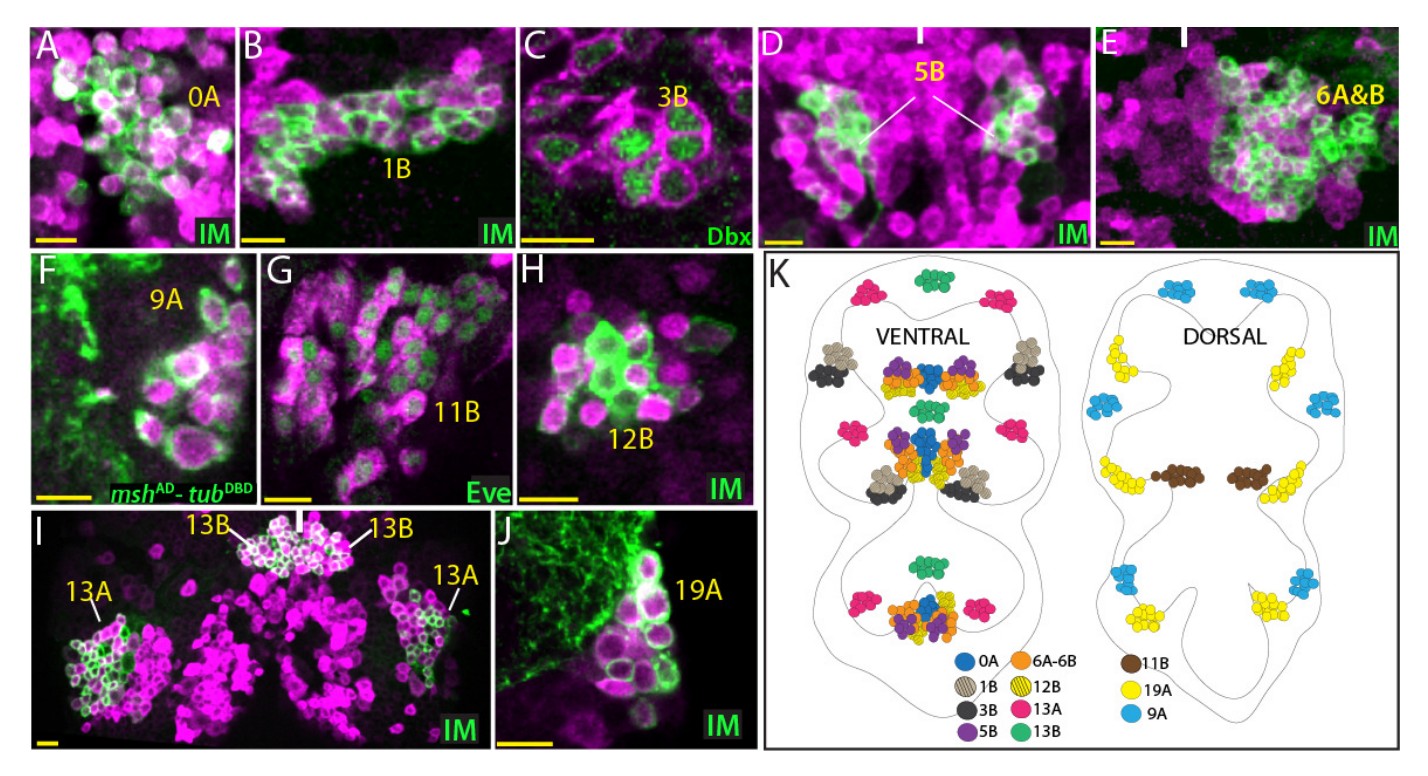

**Figure 5.** GABAergic lineages. GABA antibody (magenta) labels neurons of 0A, 1B, 5B, 6A, 6B, 9A, 11B, 12B, 13A, 13B, and 19A hemilineages in the adult VNC (**A, B, and D-J**). *gad1*-LexA reporter (magenta) labels 3B neurons in the VNC of a wandering-stage larva (**C**). NB intersected reporter immortalization (IM) used to visualize 0A, 1B, 5B, 6A, 6B, 12B, 13A, 13B, and 19A hemilineages (**A, B, D, E, H, I and J**). The GAL4 lines used for these immortalization experiments are listed in *Table 1*. Dbx, *msh*AD-*tub*DBD, and Eve used to identify neurons of 3B, 9A, and 11B, respectively (**C, F, and G**). (**K**) The location of GABAergic hemilineages shown schematically via color-coded map. Left and right images represent ventral and dorsal halves of the VNC, respectively. Only thoracic lineages are shown. All images are from a single confocal stack except the image in (**E**), which is a maximum projection of 20 optical slices (0.5 micron each). See *Figure 5—figure supplement 1* for individual channels. Solid lines indicate midline. Scale bar is 10 microns.
DOI: https://doi.org/10.7554/eLife.43701.009

The following figure supplement is available for figure 5:

**Figure supplement 1.** GABAergic hemilineages.
DOI: https://doi.org/10.7554/eLife.43701.010

reporter marks immature Dbx+ 3B neurons during larval stages (*Figure 5C*). Additionally, we used a GAL4 driver, SS20872, which marks a subset of 3B neurons and showed that gad1-LexA labels these neurons (*Figure 5—figure supplement 1C*). In summary, our results indicate that 3B neurons are GABAergic.

Neurons of the 5B cluster are located in the ventral side of the nerve cord and have complex projections (*Harris et al., 2015*). We visualized 5B adult neurons via NB intersected reporter immortalization (*Table 1*). The 5B cluster is composed of $33 \pm 3$ neurons (n = 6) in adult flies. The GABA antibody stains all of these neurons, while the ChAT antibody fails to do so (*Figure 5D*; *Figure 5—figure supplement 1D*). Thus, 5B neurons are GABAergic.

We immortalized R45D04-GAL4 reporter expression in the NB5-2 progeny to visualize the 6A and 6B clusters. Since 6A and 6B neurons intermingle, we failed to separate them. Nevertheless, we found that the entire postembryonic progeny of NB5-2, 6A and 6B neurons, are GABAergic (*Figure 5E*). We further confirmed these by findings 6A neurons with 96A08-GAL4 and 6B neurons with Vestigial (Vg) or Engrailed (En) expression and showed that both 6A and 6B neurons are GABAergic (*Figure 5—figure supplement 1E* and not shown). Based on one quantifiable sample obtained from NB intersected reporter immortalization, we detected 51 6A neurons (En negative) and 42 6B neurons (En positive) in a T1 hemisegment.

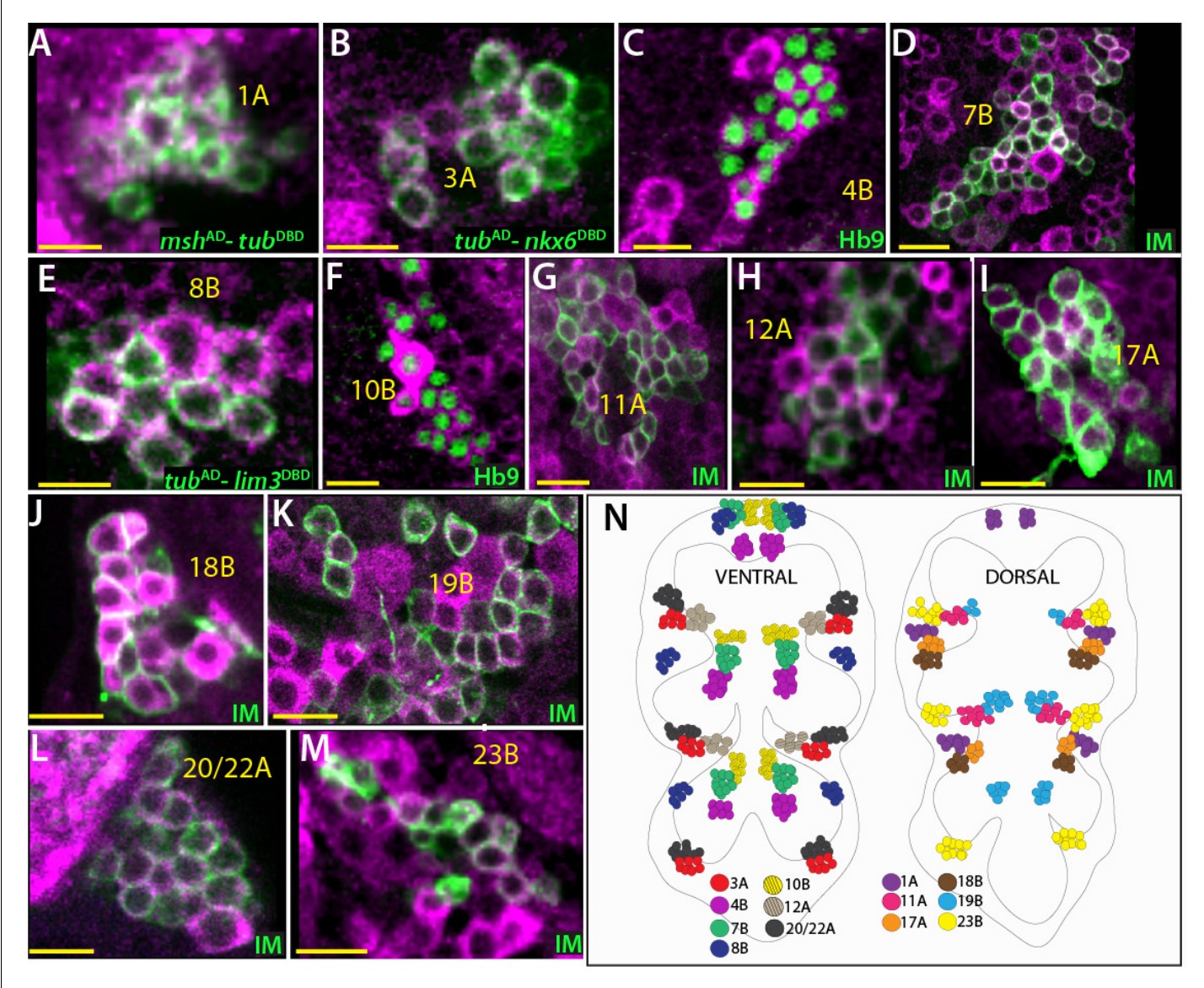

**Figure 6.** Cholinergic lineages. ChAT antibody (magenta) labels neurons of 1A, 3A, 4B, 7B, 8B, 11A, 10B, 12A, 17A, 18B, 19B, 23B and 20/22A hemilineages in the adult VNC (A–M). NB intersected reporter immortalization (IM) used to visualize the neurons of 7B, 11A, 12A, 17A, 18B, 19B, 23B and 20/22A hemilineages (D, F, and I-M). The GAL4 lines used for these immortalization experiments are listed in *Table 1*. $msh^{AD}$-$tub^{DBD}$, $tub^{AD}$-$nkx6^{DBD}$, $tub^{AD}$-$lim3^{DBD}$ used to identify neurons of 1A, 3A, and 8B, respectively (A, B, and E). Hb9 expression used to mark 4B and 10B neurons (C and G). (N) The location of cholinergic hemilineages shown schematically via color-coded map. Left and right images represent ventral and dorsal halves of the VNC, respectively. Only thoracic lineages are shown. All images are from a single confocal stack. See *Figure 6—figure supplement 1* for individual channels. Scale bar is 10 microns.

DOI: https://doi.org/10.7554/eLife.43701.011

The following figure supplement is available for figure 6:

**Figure supplement 1.** Cholinergic hemilineages.
DOI: https://doi.org/10.7554/eLife.43701.012

The 9A neurons are the major neuronal cluster that arises from NB3-5 and innervate the leg neuropil (*Harris et al., 2015*; *Lacin and Truman, 2016*). We identified 9A neurons based on Msh expression. We also used $msh^{AD}$-$tub^{DBD}$ reporter to visualize these neurons. We counted 73 ± 3 (n = 4) 9A neurons per cluster and all appeared to be immunopositive for GABA but not ChAT (*Figure 5F*; *Figure 5—figure supplement 1F*). Moreover, we failed to detect any 9A neurons with split GAL4

combination of $msh^{AD}$-$ChAT^{DBD}$ or $msh^{AD}$-$VGlut^{DBD}$ indicating the absence of cholinergic or glutamatergic neurons in 9A hemilineage (data not shown). Thus, the 9A neurons use GABA as their neurotransmitter.

Neurons of the 11B cluster are located on the dorsal surface of the nerve cord and marked by Eve expression (*Figure 5G*). In addition to Eve, we used reporter immortalization in NB7-2 progeny to identify 11B neurons (*Table 1*). We detected 68 ± 4 (n = 3) 11B neurons in T1 hemisegments. In T2 and T3, the number of 11B neurons was dramatically reduced to 7 and 2 neurons, respectively (n = 2 for each). Both GABA immunostaining and *gad1*-lexA reporter mark 11B neurons while *ChAT*-LexA fails to mark any 11B neurons. (*Figure 5G*; *Figure 5—figure supplement 1G*). Thus, 11B neurons are GABAergic.

The 12B neurons innervate the leg neuropil, and they are located medially relative to their sibling 12A lineage (*Harris et al., 2015*). We visualized 12A and 12B adult neurons via reporter immortalization in the progeny of NB6-1 (*Table 1*) and identified 12B neurons based on their lateral location. We also used the expression of Nkx6 and Nmr1, which are expressed in 12B neurons in a partially overlapping manner (*Figure 5—figure supplement 1H*). We detected 67 ± 7 (n = 4) 12B neurons per cluster and found that 12B neurons were marked with GABA antibody but none showed ChAT expression (*Figure 5H*; *Figure 5—figure supplement 1H*). Thus, our results indicate that the 12B neurons are GABAergic.

The 13A and 13B clusters arise from NB4-2 and innervate the leg neuropil ipsilaterally and contralaterally, respectively (*Harris et al., 2015*; *Lacin and Truman, 2016*). We visualized these sibling lineages via NB intersected reporter immortalization (*Table 1*). The 13A neurons are located laterally compared to the medially located 13B neurons (*Figure 5I*). The 13A cluster is composed of 67 ± 6 neurons (n = 3) and around half of them express Dbx; the 13B cluster is composed of 47 ± 1 neurons (n = 4) and most express Dichaete (*Figure 5—figure supplement 1I*). We found that both the 13A and 13B clusters were marked with GABA, but not ChAT (*Figure 5I*; *Figure 5—figure supplement 1I*). Of note, we found that 13B neurons exhibit higher GABA levels than 13A neurons across several animals (n = 4) (*Figure 5I*; *Figure 5—figure supplement 1I*). In conclusion, both the 13A and 13B hemilineages are GABAergic.

Neurons of the 19A cluster are located dorso-laterally in the nerve cord and innervate the leg neuropil (*Harris et al., 2015*). We visualized them via reporter immortalization in the NB6-2 progeny (*Table 1*) and identified them based on their more lateral location compared to 19B neurons. We detected 62 ± 4 (n = 4) 19A neurons per cluster and most express Dbx (*Figure 5—figure supplement 1J*). Based on immunostainings, we found that 19A neurons were marked by GABA, and we did not detect any 19A neuron expressing ChAT, indicating that the 19A neurons are GABAergic (*Figure 5J*; *Figure 5—figure supplement 1J*).

In conclusion, we identified 12 GABAergic hemilineages. Half of them (1B, 9A, 12A, 13A, 13B, and 19A) are composed of interneurons innervating the leg neuropil. The other half either exhibit complex projection patterns across the VNC or innervate the flight related neuropils. Like glutamatergic lineages, GABAergic lineages appeared to be composed of purely GABAergic neurons. In some lineages (e.g. 9A, 12B, 13A, 13B, and 19A), we detected occasional GABA negative cells. We think this is mainly due to the inability of GABA antibody to work robustly.

## Cholinergic lineages

Acetylcholine serves as a fast-acting excitatory neurotransmitter in the *Drosophila* CNS (*Lee and O'Dowd, 1999*). To identify cholinergic neurons, we used genetic reporters (*ChAT*$^{GAL4}$, *ChAT*$^{AD}$ and *ChAT*$^{LexA}$ (*Diao et al., 2015*)) and a ChAT specific antibody (*Takagawa and Salvaterra, 1996*). Identifying the cholinergic fate was challenging because unlike VGlut and gad1 reporters, the ChAT reporters labeled many non-cholinergic neurons in addition to cholinergic neurons (*Figure 7—figure supplement 1*; also see below). Thus, we relied on ChAT antibody staining and identified 14 cholinergic hemilineages: 1A, 3A, 4B, 7B, 8B, 10B, 11A, 12A, 17A, 18B, 19B, 23B, 20/22A (*Figure 6N*). Importantly, all of the clusters that showed ChAT immunoreactivity were positive for the ChAT reporters (not shown).

We used NB1-2 intersected reporter immortalization (*Table 1*), Msh protein expression, and the Msh$^{AD}$-Chat$^{DBD}$ split GAL4 combination to visualize the 1A neurons. The 1A hemilineage contain 59 ± 3 neurons (n = 4) in segments T1 and T2 and 40 ± 1 neurons (n = 2) in the T3 segment. We observed ChAT expression in 1A neurons and failed to detect GABA expression (*Figure 6A*;

*Figure 6—figure supplement 1A*). Moreover, msh$^{AD}$-VGlut$^{DBD}$ combination did not mark any 1A neurons, indicating the absence of glutamatergic 1A neurons (*Figure 3A*). Hence, the neurons of the 1A cluster are cholinergic.

As mentioned above, we used NB7-1 intersected reporter immortalization (*Table 1*) together with Nkx6 expression to mark the 3A neurons. 3A neurons innervate the leg neuropil ipsilaterally (*Harris et al., 2015*). We detected 75 ± 4 (n = 3) 3A neurons per cluster in T1 segments, where it was feasible to quantify this lineage reliably. We expect T2-T3 segments to have a similar number of 3A neurons since leg-neuropil related lineages show little, if any, segmental differences. Based on transmitter immunostaining we found that Nkx6-expressing 3A neurons were positive for ChAT expression and negative for GABA expression (*Figure 6B*; *Figure 6—figure supplement 1B*; *Figure 5—figure supplement 1C*). However, we detected a few Nkx6$^+$/GABA$^+$ neurons which also express Dbx, an early marker for GABAergic 3B neurons (*Figure 6—figure supplement 1B*). We were not able to assign these cells to a specific hemilineage in NB7-1 progeny. Nevertheless, our results suggest that NB7-1 progeny is composed of cholinergic 3A neurons and GABAergic 3B neurons.

We used Hb9 or Hb9-GAL4 expression to identify the 4B neurons, which are located ventrally and innervate the leg neuropil (*Harris, 2013*). We detected ChAT expression in all 4B neurons (44 ± 1 neurons per cluster n = 4) and failed to detect GABA or VGlut immunostaining in them (*Figure 6C*; *Figure 6—figure supplement 1C*). Thus, we concluded that the 4B neurons are cholinergic.

NB3-2 generates only 7B neurons, as 7A neurons are eliminated by apoptosis soon after birth. We visualized 7B neurons via NB3-2 intersected reporter immortalization (*Table 1*). Axon 7B neuronal projections show segmental differences although cell number per cluster varied little across thoracic segments. We found that 7B neurons (60 ± 6 per cluster, n = 5) express ChAT and lack GABA and VGlut immunoreactivity, and thus they are cholinergic (*Figure 6D*; *Figure 6—figure supplement 1D*).

We used NB3-3 intersected reporter immortalization in combination with Lim3 expression to identify the 8B neurons (*Table 1*). We detected 54 ± 3 (n = 4) 8B neurons per cluster, and they were labelled by ChAT antibody but not GABA (*Figure 6E*; *Figure 6—figure supplement 1E*). Thus, our results indicate 8B neurons are cholinergic. Of note, we found that Acj6 is expressed in all 8B neurons in T1 segments while the number of Acj6 expressing 8B neurons in T2-T3 segments is significantly reduced (not shown). This discrepancy suggests that Acj6 might regulate 8B axonal projections, which show segment specific differences.

NB2-2 generates only 10B neurons. We used NB2-2 intersected reporter immortalization in combination with Hb9 and Lim3 expression to identify the 10B neurons (*Table 1*). We detected 57 ± 3 (n = 3) 10B neurons per cluster. Based on the antibody staining against ChAT, GABA, and VGlut, we found that 10 B neurons are cholinergic (*Figure 6F*; *Figure 6—figure supplement 1F*).

NB7-2 generates the 11A neurons in addition to the Eve$^+$ GABA$^+$ 11B neurons (*Lacin et al., 2014*; *Lacin and Truman, 2016*). We visualized the entire progeny of NB7-2 via reporter immortalization and assumed neurons lacking GABA and Eve staining were the 11A neurons (*Table 1*). 11A neurons are present only in the T1 and T2 segments. We detected 46 ± 2 (n = 4) 11A neurons per hemisegment and almost all were stained with ChAT antibody (*Figure 6G*; *Figure 6—figure supplement 1G*). In addition, we used tub$^{AD}$-nkx6$^{DBD}$ split combination, which marks half of the 11A neurons and confirmed by ChAT immunostaining that the 11A neurons are cholinergic (*Figure 6—figure supplement 1G*).

Neurons of the 12A and 12B clusters derive from NB6-1 (*Lacin and Truman, 2016*). We visualized both hemilineages via NB intersected reporter immortalization (*Table 1*) and identified 12A neurons based on their lateral position compared to GABA$^+$12B neurons (*Harris et al., 2015*). 12A hemilineage is present in T1 and T2 and absent in T3 segments; we quantified only T1 segments and counted 55 ± 6 (n = 4) 12A neurons per cluster. ChAT marked 12A neurons while GABA and VGlut did not label 12A neurons (*Figure 6H*; *Figure 6—figure supplement 1H*). Thus, our results indicate that the 12A neurons are cholinergic.

NB2-5 generates only the 17A neurons since the 17B neurons are eliminated by apoptosis (*Truman et al., 2010*; *Lacin and Truman, 2016*). The 17A neurons are located on the dorsal surface of the nerve cord and are found only in the T2 and T3 segments (*Harris et al., 2015*). We visualized 17B neurons via NB2-5 intersected reporter immortalization (*Table 1*) or Islet expression. We

detected 54 ± 3 (n = 2) and 32 ± 4 (n = 3) 17B neurons in the T2 and T3 segments, respectively, and all of them expressed ChAT protein and lacked GABA staining (*Figure 6I*; *Figure 6—figure supplement 1I*). Thus, neurons in the 17A hemilineage are cholinergic.

The 18B neurons arise from NB2-4 as the only surviving hemilineage, and they are located on the dorsal part of the nerve cord, just posterior to 17A neurons (*Harris, 2013*; *Harris et al., 2015*; *Lacin and Truman, 2016*). Like the 17A neurons, they are located only in the T2 and T3 segments. NB2-4 intersected reporter immortalization (*Table 1*) marked 28 ± 2 (n = 3) and 18 (n = 1) 18B neurons per cluster in the T2 and T3 segments, respectively. We found that 18B neurons express ChAT and lack GABA. Thus, our results identify 18B neurons as cholinergic (*Figure 6J*; *Figure 6—figure supplement 1J*).

Neurons of the 19B hemilineage are located on the dorsal surface of the VNC and arise from NB6-2 (*Harris et al., 2015*; *Lacin and Truman, 2016*). We identified 19B neurons in the NB intersected reporter immortalization clones (*Table 1*) based on their medial location compared to their sibling GABAergic 19A neurons, which are located at the lateral edge of the VNC (*Table 1*). We detected 63 ± 1 (n = 2) and 27 (n = 1) 19B neurons per T2 and T3 hemisegments, respectively. The number of 19B neurons is greatly reduced to 8 ± 1 neurons (n = 2) per T1 cluster. We found that the ChAT but not GABA antibody marks 19B neurons, indicating that the 19B neurons are cholinergic (*Figure 6K*; *Figure 6—figure supplement 1K*).

NB5-4 and NB5-7 generate the 20A and 22A hemilineages respectively (*Lacin and Truman, 2016*). Neurons from these hemilineages innervate the leg neuropil (*Harris et al., 2015*). We co-visualized both hemilineages by NB intersected reporter immortalization (*Table 1*). Since these hemilineages are located in the same region and intermixed, we could not separate them. In total, we detected 110 ± 6 (n = 5) 20A/22A neurons per cluster. All of them appeared to be cholinergic as they expressed ChAT but not GABA (*Figure 6L*; *Figure 6—figure supplement 1L*). We also detected a few glutamatergic but ChAT negative neurons, which are presumably the surviving motor neurons from the 'B' hemilineages of 20/22 lineages (not shown).

23B is the only surviving hemilineage from NB7-4; these neurons innervate the leg neuropil (*Harris et al., 2015*; *Lacin and Truman, 2016*). We visualized them via NB7-4 intersected reporter immortalization (*Table 1*) in addition to Acj6 expression. We detected 61 ± 3 23B neurons per cluster (n = 6). ChAT antibody marked all of the 23B neurons while GABA and VGlut specific reagents failed to label the 23B neurons (*Figure 6M*; *Figure 6—figure supplement 1M*; not shown). We concluded that 23B neurons are cholinergic.

In conclusion, we identified 14 cholinergic hemilineages. Among them, 3A, 4B, 20/22A, and 23B heavily innervate the leg neuropil. The rest project to the flight related neuropils or facilitate the communication between different neuropils. Similar to glutamatergic and GABAergic lineages, these 14 hemilineages appeared to be composed of purely cholinergic neurons. We were unable to ascertain this for a few lineages (1A, 8B, 11A, and 19B), which contained occasional ChAT negative cells. We believe this is due to the inability of the ChAT antibody to label cell bodies strongly in a reproducible manner, but it is possible that these lineages may contain a few neurons that are not cholinergic. Additionally, we found that ChAT reporters marked the entirety of these lineages, thus supporting the former possibility, but with the caveat that ChAT reporters label a subset of non-cholinergic neurons. Of note, due to the limited availability of the VGlut antibody, we only tested a few non-glutamatergic lineages for the absence of VGlut protein.

## Co-expression of neurotransmitter

Our observation that ChAT reporters mark noncholinergic neurons (e.g. 14A neurons, *Figure 1B*) made us investigate in more detail if these and other neurons use more than one fast acting neurotransmitter. First, we used the split GAL4 combination of *gad1*[AD] and *VGlut*[DBD] to test if any neuron co-expresses GABA and glutamate. We did not find any neuron in either the VNC or brain that reproducibly expressed UAS-linked reporter gene with this combination, indicating the absence of such neurons (*Figure 7A,B*). Moreover, antibody staining revealed that *gad1* and *VGlut* reporters marked only GABA[+] or VGlut[+] neurons, respectively (*Figure 7—figure supplement 1A,B*). Thus, our findings indicate that no neurons use both GABA and glutamate neurotransmitters.

Next, we tested if ChAT is expressed by GABAergic and glutamatergic neurons, which were visualized by *gad1* and *VGlut* reporters, respectively. We failed to and detect any significant ChAT staining in these neurons. Since ChAT antibody does not produce robust signals reproducibly, we utilized

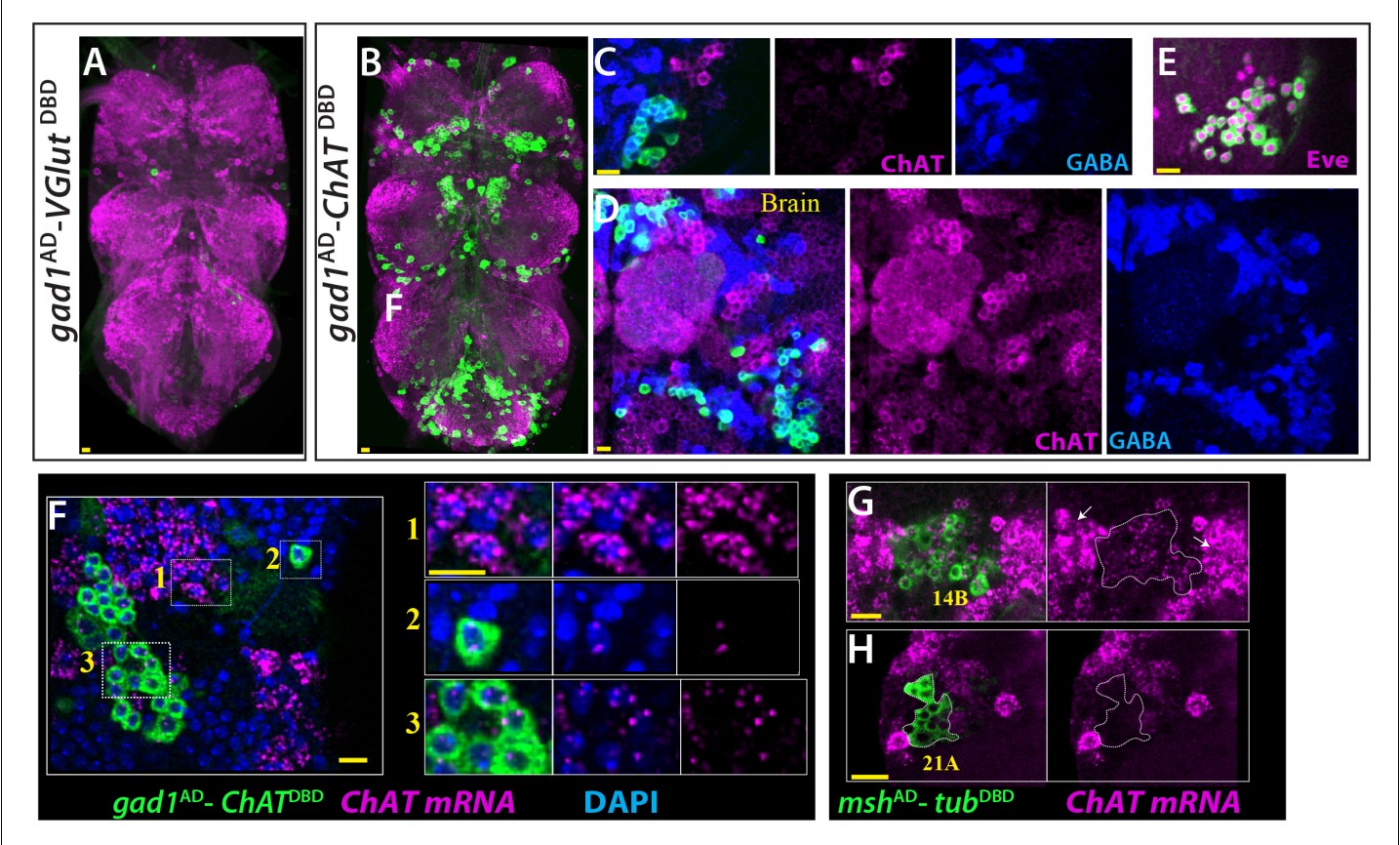

**Figure 7.** ChAT is transcribed in many Glutamatergic and GABAergic neurons. (**A**) *gad1*[AD]-*VGlut*[DBD] does not mark any neuron reproducibly in the adult VNC. (**B–D**) *gad1*[AD]-*ChAT*[DBD] mark several clusters of neurons in both the VNC (**B, C**) and brain (**D**). These neurons are positive for GABA immunoreactivity (blue) and negative for ChAT immunoreactivity (magenta). (**E**) The dorsal VNC neurons marked with *gad1*[AD]-*ChAT*[DBD] are members of 11B hemilineage, which express Eve (Magenta). (**F**) *gad1*[AD]-*ChAT*[DBD] marked neurons (inset 2, 3) contains low levels of *ChAT* mRNA (Magenta), which are mostly restricted to the nucleus while cholinergic neurons (inset 1) contains copious amount of *ChAT* mRNA, which are enriched in the cytoplasm. (**G–H**) *msh*[AD]-*tub*[DBD] used to mark glutamatergic14B and 21A neurons. (**G**) 14A neurons (outlined with white dashes) express low levels *ChAT* mRNA compared to cholinergic neurons with high ChAT expression (arrows). (**H**) Glutamatergic 21A neurons completely lack *ChAT* mRNA. Images in (**A, B**) are maximum confocal projections; the rest of the images are from single confocal plane. Scale bar is 10 microns.

DOI: https://doi.org/10.7554/eLife.43701.013

The following figure supplement is available for figure 7:

**Figure supplement 1.** Characterization of neurotransmitter-specific genetic reporters.

DOI: https://doi.org/10.7554/eLife.43701.014

genetic tools reporting ChAT expression (*Diao et al., 2015*) to test if cholinergic neurons use other major neurotransmitters. To our surprise, we found that *ChAT*-GAL4 marked many neurons in both the VNC and brain that were also labeled by GABA or VGLUT antibodies (*Figure 7—figure supplement 1C*). We focused on the GABAergic set of ChAT-GAL4 marked neurons by visualizing them via *gad1*[AD]-*ChAT*[DBD] (*Figure 7B*). We found that most of these neurons are members of the 5B and 11B hemilineages of the VNC and other lineages in the brain (*Figure 7C–E*). Antibody staining showed that GABA marks these neurons strongly while the ChAT antibody did not label any of them. To see if these neurons express ChAT transcript, we performed a modified version of RNA in situ hybridization, which employs a chain reaction and can reveal transcripts with low copy numbers (*Choi et al., 2018*).We found that these ChAT-immunonegative neurons indeed express ChAT transcript, but at very low levels compared to neurons in clusters that showed ChAT immunostaining (Figure F). Most of the detected transcripts were in the nucleus, typically appearing as two transcription dots per nucleus. We interpret these as being transcription spots formed by the nascent transcripts associated with the two ChAT genes per nucleus. We rarely detected small amounts of ChAT signal

outside the nucleus (not shown). We expect that the transcripts are likely actively degraded and do not result in transcript accumulation in the cytoplasm or in ChAT protein. We also observed low levels of ChAT transcripts in glutamatergic neurons of the clusters of 14A interneurons and 15B motor neurons (Figure G, H and not shown). These neurons also showed up using the ChAT genetic reporters, but failed to show ChAT immunostaining. In conclusion, our results indicate that the *ChAT* gene is transcribed in glutamatergic and GABAergic cells of certain lineages and suggests that these transcripts do not result in functional enzyme.

## Discussion

The *Drosophila* VNC is a great model system to investigate how neurons acquire different fates, form functional circuits, and direct specific vital behaviors. To study the functional assembly of this complex tissue, we have taken a lineage-based approach to divide the VNC into developmentally and functionally related units, which are the hemilineages.

The adult VNC is made up primarily from 34 hemilineages - clusters of lineally related, segmentally repeated, postembryonic-born neurons. Previous work systematically characterized the development and neuronal morphologies of these hemilineages, and showed that neurons within a hemilineage adopt similar fates, evident from their immature axonal projection and transcription factor expression (*Truman et al., 2004*; *Truman et al., 2010*; *Lacin et al., 2014*; *Harris et al., 2015*; *Shepherd et al., 2016*). Here, we systematically mapped the neurotransmitter choice of most VNC neurons by studying all of the postembryonic hemilineages. Surprisingly, we found that neurotransmitter code in the VNC is simple in that all neurons within a hemilineage use the same neurotransmitter, thus transmitter identity is determined at the stem cell level. These results further support our earlier findings that hemilineages represent functional units, both in terms of anatomy and now neurotransmitter chemistry.

Our statement that all neurons within a hemilineage use the same neurotransmitter excludes the neurons that are born during early embryonic neurogenesis. As mentioned earlier, neurons born during this time are highly diverse and might use a different neurotransmitter than the rest of the neurons in the hemilineage. For example, both NB4-2 and NB5-2 generate glutamatergic motor neurons from their early embryonic divisions (*Landgraf et al., 1997*; *Schmid et al., 1999*; *Lacin and Truman, 2016*), but their postembryonic progenies (both A and B hemilineages) are purely GABAergic. Similarly, glutamatergic U/CQ motor neurons, which are born in early embryonic stages share the same hemilineage with the cholinergic 3A neurons (*Landgraf et al., 1997*; *Schmid et al., 1999*; *Lacin et al., 2009*). Thus, neuronal fates within a hemilineage can be dramatically different when embryonic and postembryonic neurons are compared. We believe the reason why postembryonic hemilineages in the VNC are homogenous in terms of neuronal fate is due to the expansion of particular, later-born neuronal classes as neuronal lineages became larger during evolution of more derived insects to accommodate more complex behaviors such as flight. Since all insect species have similar sets of NBs, new behaviors (e.g. flight) appear to have evolved via changes in the number of neurons generated by each NB, but not changes in the number of NBs (*Truman and Bate, 1988*). Moreover, the Notch mediated asymmetric division enabled the insect to have two distinct clonal populations of neurons (hemilineages) from a single NB. Interestingly, only 34 of 50 potential hemilineages are used in the adult fly VNC while 16 of them are eliminated by apoptosis. Thus, flies have the potential to acquire novel behaviors by simply resurrecting hemilineages that are fated to die.

At least within the thorax, the hemilineages express the same transmitter regardless of their segment of residence. This conservation was expected for the hemilineages that contribute to the leg neuropils, since neuron numbers and the projections of these cells appear similar across the different thoracic segments. On the other hand, the hemilineages innervating the dorsal, flight-related neuropils have segment specific organization and show dramatically different axonal projections depending on their segmental location. For example, 7B neurons in each segment have unique projection and appear to execute distinct behaviors (*Harris et al., 2015*). Despite these differences, 7B neurons use acetylcholine in every segment. These results show that the neurotransmitter fate is tightly linked to the lineage origin and that the segmental diversification of the 7B neurons with the evolution of the derived flight system of flies may have had to occur with this transmitter constraint.

We expect that most hemilineages in the fly brain are also homogenous in terms of neurotransmitter expression as we observed large neuronal clusters expressing the same neurotransmitter.

However, some complex brain hemilineages that have diverse neuronal populations might have different neurotransmitter expression as it was shown for the lAL lineage (*Lai et al., 2008*; *Lin et al., 2010*).

Here, we also extended our earlier transcription factor expression studies in immature neurons of larval stages into the mature neurons of the adult. We found that the expression of many transcription factors is maintained into adult stages and can be used to mark specific hemilineages in the adult. However, some transcription factors are expressed transiently during development. For example, Dbx marks many immature 3B neurons in the larva (*Lacin et al., 2014*), but its expression disappears in these neurons after pupa formation (unpublished data). From the expression analysis of the limited number of transcription factors studied here, we did not find any factor that specifically marked all neurons of a specific neurotransmitter type. However, we identified a few transcription factors, whose expression tightly correlated with the neurotransmitter fate. For example, Dbx expression is restricted to GABAergic neurons, even though Dbx does not appear to promote the GABA fate by itself as GABAergic fate is unaltered in response to Dbx loss or misexpression (unpublished data). Similarly, we found that Unc-4 expression is restricted to cholinergic lineages among the postembryonic lineages in the VNC; however, in the brain Unc-4 is expressed in glutamatergic lineages in addition to cholinergic lineages, suggesting that different parts of the CNS might use the same transcription factor for different fates (unpublished data) via utilizing different cofactors. Supporting this, we found that none of the GABAergic lineages in the VNC are marked with Lim3, which was shown to be expressed in most GABAergic neurons of the fly optic lobe and required for their GABAergic identity (*Konstantinides et al., 2018*). The reverse scenario where neurons acquire the same neurotransmitter identity via different transcriptional regulatory networks is also commonly observed. For example in the *C. elegans* nervous system, distinct combinations of 13 transcription factors are responsible for VGlut expression in 25 different glutamatergic neuron classes (*Serrano-Saiz et al., 2013*). Thus, transcription factors act together combinatorially rather than individually to specify neurotransmitter fate.

Our study did not find any correlation between the Notch state of the neurons and neurotransmitter identity, an observation made in the optic lobe (*Konstantinides et al., 2018*). We found that any neurotransmitter type can be observed in both 'A' and 'B' hemilineages. We also noted that NB4-2 (progenitor of 13A/B) and NB5-2 (progenitor of 6A/B) are the only two NBs in which both the 'A' and 'B' hemilineages use the same neurotransmitter, GABA.

Interestingly, the neurotransmitter pattern of *Drosophila* hemilineages appears to be conserved in other insect species. For example, based on location and morphology, we deduced that sibling 'K$_l$' and 'K$_m$' GABAergic clusters of the moth, Manduca Sexta, are homologous to the 13A and 13B hemilineages, respectively, and the 'M' cluster is likely homologous to the 6A, 6B, and 5B neurons, which form a large cluster GABAergic in the posterior thoracic ganglia of the fly (*Witten and Truman, 1991*). Similar GABAergic clusters were also observed in the nerve cords of grasshopper (*O'Dell and Watkins, 1988*) and silverfish (unpublished data, HL and JWT). Interestingly, like the Drosophila VNC, the grasshopper nerve cord contains two clusters of En$^+$GABA$^+$ neurons, named 'A' and 'B' groups, which are likely homologous to 0A and 6B neurons, respectively (*Siegler et al., 2001*).

Unexpectedly, we detected ChAT transcripts in many GABAergic and glutamatergic neurons, most of which are members of lineages 5B and 11B (GABAergic) and 14A and 15B (glutamatergic). The low levels of ChAT transcripts and the lack of ChAT immunostainings in these cells suggested that ChAT transcripts are actively degraded and not translated. It is possible that these neurons produce acetylcholine but only in certain conditions for example during development or under stress. Indeed, neurotransmitter switching has been observed in many neurons of vertebrates, however, most of these switches involve aminergic neurotransmitters (*Spitzer, 2015*).

Another possibility for the presence of ChAT transcripts in noncholinergic neurons is that it is a remnant of a neurotransmitter switch that might have happened during evolution. 15B neurons are a good candidate for such a possibility. 15B motor neurons are glutamatergic like all other fly motor neurons, while all vertebrate and some invertebrate (e.g, *C. elegans* and *Aplysia*) motor neurons are cholinergic (reviewed in *Fieber, 2017*), raising the possibility that motor neurons of the common ancestor used acetylcholine. Thus, the ChAT expression in 15B motor neurons might be a vestige from the cholinergic ancestry of motor neurons.

# Materials and methods

**Key resources table**

| Reagent type (species) or resource | Designation | Source or reference | Identifiers | Additional information |
|---|---|---|---|---|
| Antibody | Mouse anti-ChaT monoclonal | DSHB | ChAT4B1 | 1:50 diltuion |
| Antibody | Rabbit anti-GABA polyclonal | Sigma | A2052 | 1:1000 dilution |
| Antibody | Mouse anti-GABA monoclonal | Sigma | A0310 | 1:1000 dilution |
| Antibody | Rabbit antiVGlut polyclonal | *Daniels et al., 2004* | | 1:10000 dilution |
| Antibody | Mouse anti-BRP monoclonal | DSHB | nc82 | 1:50 diltuion |
| Antibody | Chicken anti-GFP polyclonal | Life Tech. | A10262 | 1:1000 dilution |
| Antibody | Rabbit anti-DsRED polyclonal | TAKARA | 632496 | 1:500 dilution |
| Antibody | Rabbit anti-Vg polyclonal | Gift from Sean Carroll | | 1:500 dilution |
| Antibody | Guinea pig anti-Ems polyclonal | Gift from U. Walldorf | | 1:300 dilution |
| Antibody | Rabbit anti-Msh polyclonal | Gift from Chris Doe | | 1:500 dilution |
| Antibody | Rabbit anti-Unc-4 polyclonal | Gift from James Skeath | | 1:1000 dilution |
| Antibody | Guinea pig anti-Dbx polyclonal | Gift from James Skeath | | 1:1000 dilution |
| Antibody | Rabbit anti-HB9 polyclonal | Gift from James Skeath | | 1:1000 dilution |
| Antibody | Guinea pig anti-HB9 polyclonal | Gift from James Skeath | | 1:1000 dilution |
| Antibody | Rat anti-Nkx6 polyclonal | Gift from James Skeath | | 1:1000 dilution |
| Antibody | Guinea pig anti-Lim3 polyclonal | Gift from James Skeath | | 1:1000 dilution |
| Antibody | Rat anti-Islet polyclonal | Gift from James Skeath | | 1:1000 dilution |
| Antibody | Rabbit anti-Dichaete polyclonal | Gift from James Skeath | | 1:1000 dilution |
| Antibody | Rabbit anti-Nmr1 polyclonal | Gift from James Skeath | | 1:1000 dilution |
| Antibody | Mouse anti-En monoclonal | DSHB | 4D9 | 1:5 dilution |
| Antibody | Mouse anti-Acj6 monoclonal | DSHB | Acj6 | 1:100 dilution |
| Antibody | Mouse anti-Eve monoclonal | DSHB | 3C10 | 1:25 dilution |
| Antibody | goat anti-rabbit Alexa Fluor 488 | Life Technologies | A-11034 | 1:500 dilution |
| Antibody | goat anti-rabbit Alexa Fluor 568 | Life Technologies | A-11011 | 1:500 dilution |
| Antibody | goat anti-rabbit Alexa Fluor 633 | Life Technologies | A-21070 | 1:500 dilution |

*Continued on next page*

*Continued*

| Reagent type (species) or resource | Designation | Source or reference | Identifiers | Additional information |
|---|---|---|---|---|
| Antibody | goat anti-rabbit Alexa Fluor 405 | Life Technologies | A-31556 | 1:500 dilution |
| Antibody | goat anti-chicken Alexa Fluor 488 | Life Technologies | A-11039 | 1:500 dilution |
| Antibody | goat anti-mouse Alexa Fluor 488 | Life Technologies | A-11001 | 1:500 dilution |
| Antibody | goat anti-mouse Alexa Fluor 568 | Life Technologies | A-11004 | 1:500 dilution |
| Antibody | goat anti-mouse Alexa Fluor 633 | Life Technologies | A-21050 | 1:500 dilution |
| Antibody | goat anti-mouse Alexa Fluor 405 | Life Technologies | A-31553 | 1:500 dilution |
| Antibody | goat anti-guinea pig Alexa Fluor 488 | Life Technologies | A-11073 | 1:500 dilution |
| Antibody | goat anti-guinea pig Alexa Fluor 568 | Life Technologies | A-11075 | 1:500 dilution |
| Antibody | goat anti-guinea pig Alexa Fluor 633 | Life Technologies | A-21105 | 1:500 dilution |
| Antibody | goat anti-rat Alexa Fluor 488 | Life Technologies | A-11006 | 1:500 dilution |
| Antibody | goat anti-rat Alexa Fluor568 | Life Technologies | A-11077 | 1:500 dilution |
| Genetic reagent (*D. melanogaster*) | VGlut-GAL4 | Bloomington Drosophila Stock Center | RRID:BDSC_60312 | |
| Genetic reagent (*D. melanogaster*) | VGlutDBD | Bloomington Drosophila Stock Center | RRID:BDSC_60313 | |
| Genetic reagent (*D. melanogaster*) | VGlut-LexA | Bloomington Drosophila Stock Center | RRID:BDSC_60314 | |
| Genetic reagent (*D. melanogaster*) | ChAT-GAL4 | Bloomington Drosophila Stock Center | RRID:BDSC_60317 | |
| Genetic reagent (*D. melanogaster*) | ChAT-DBD | Bloomington Drosophila Stock Center | RRID:BDSC_60318 | |
| Genetic reagent (*D. melanogaster*) | ChAT-LexA | Bloomington Drosophila Stock Center | RRID:BDSC_60319 | |
| Genetic reagent (*D. melanogaster*) | gad1-AD | Bloomington Drosophila Stock Center | RRID:BDSC_60322 | |
| Genetic reagent (*D. melanogaster*) | gad1-LexA | Bloomington Drosophila Stock Center | RRID:BDSC_60324 | |
| Genetic reagent (*D. melanogaster*) | Tub-AD | Bloomington Drosophila Stock Center | RRID:BDSC_60295 | |
| Genetic reagent (*D. melanogaster*) | Tub-DBD | Bloomington Drosophila Stock Center | RRID:BDSC_60298 | |
| Genetic reagent (*D. melanogaster*) | SS20873 | This study | | |

*Continued on next page*

*Continued*

| Reagent type (species) or resource | Designation | Source or reference | Identifiers | Additional information |
|---|---|---|---|---|
| Chemical compound, drug | Paraformaldehyde | EMS | 15713 | |
| Chemical compound, drug | Vectashield | Vector | H-1000 | |
| Chemical compound, drug | Prolong Diamond | Molecular Probes | P36961 | |
| Recombinant DNA reagent | pBS-KS-attB2-SA(2)-T2A-Gal4DBD-Hsp70 | addgene | RRID:Addgene_62904 | |
| Recombinant DNA reagent | pBS-KS-attB2-SA(1)-T2A-Gal4DBD-Hsp70 | addgene | RRID:Addgene_62903 | |
| Recombinant DNA reagent | pBS-KS-attB2-SA(0)-T2A-Gal4DBD-Hsp70 | addgene | RRID:Addgene_62902 | |
| Recombinant DNA reagent | pBS-KS-attB2-SA(2)-T2A-p65AD-Hsp70 | addgene | RRID:Addgene_62915 | |
| Recombinant DNA reagent | pBS-KS-attB2-SA(1)-T2A-p65AD-Hsp70 | addgene | RRID:Addgene_62914 | |
| Recombinant DNA reagent | pBS-KS-attB2-SA(0)-T2A-p65AD-Hsp70 | addgene | RRID:Addgene_62912 | |

## Immunochemistry

For a better quality of neurotransmitter staining, we used pharate or newly emerged female adult flies. Nervous systems were dissected in ice cold phosphate buffered saline (PBS) and fixed with 2% paraformaldehyde in PBS for an hour at room temperature and then washed several times in PBS-TX (PBS with 1% Triton-X100) for a total 20 min. For VGlut, Bouin's fixative was used as described (*Daniels et al., 2004*). Tissues were incubated with primary antibodies for four hours at room temperature or overnight 4°C. After three to four rinses with PBS-TX to remove the primary antisera, tissues were washed with PBS-TX for an hour. After wash, tissues were incubated with secondary antibodies for two hours at room temperature or overnight at 4°C. Tissues were washed again with PBS-TX for an hour after several rinsing and mounted in Vectashield or in DPX (*Truman et al., 2004*) after dehydration through an ethanol series and clearing in xylene.

## NB intersected reporter immortalization

Flies with the genotype of dpn >KDRT-stop-KDRT>CrePEST; Act5c > loxP-stop-loxP>LexA::p65, lex-Aop-myrGFP; UAS-KD are crossed with the NB specific GAL4 lines (*Table 1*) to immortalize reporter expression in the progeny of NBs (*Awasaki et al., 2014*; *Lacin and Truman, 2016*). Note that the GAL4 lines used for reporter immortalization may mark other lineages; the lineage of interest was identified based on location, axonal projection and molecular markers.

## Multi-color flip out generation

GAL4 lines of interest were crossed with the MCFO-1 line (hsFlp2::PEST;; HA_V5_FLAG; (*Nern et al., 2015*). Newly hatched larvae from this cross were heat-shocked in a 37°C water bath for 15–30 min to generate NB lineage clones.

## Drosophila husbandry

Stocks are maintained in the standard cornmeal-based fly food in a 25C incubator under humidity control. The following fly lines were obtained from Bloomington Drosophila Stock Center (BDSC):

VGlut-GAL4 (w[*]; Mi{Trojan-GAL4.2}VGlut[MI04979-TG4.2]/CyO)

VGlut[DBD] (w[*]; Mi{Trojan-GAL4.2}VGlut[MI04979-TG4.2]/CyO)
VGlut-LexA    (w[*];    Mi{Trojan-lexA:QFAD.2}VGlut[MI04979-TlexA:QFAD.2]/CyO    P{Dfd-GMR-nvYFP}2
ChAT-GAL4 (w[*]; Mi{Trojan-GAL4.0}ChAT[MI04508-TG4.0]/TM6B, Tb[1])
ChAT[DBD] (w[*]; l(2)*[*]/CyO; Mi{Trojan-GAL4DBD.0}ChAT[MI04508-TG4DBD.0]/TM3, Sb[1])
ChAT-LexA (w[*]; Mi{Trojan-lexA:QFAD.0}ChAT[MI04508-TlexA:QFAD.0]/TM6B, Tb) gad1[AD] (y[1] w[*]; Mi{Trojan-p65AD.2}Gad1[MI09277-Tp65AD.2]/TM6B, Tb[1]) gad1-LexA (w[*]; Mi{Trojan-lexA:QFAD.2}Gad1[MI09277-TlexA:QFAD.2]/TM6B, Tb[1])- (*Diao et al., 2015*)
Tub[AD] (y[1] w[*]; P{w[+mC]=Tub-dVP16AD.D}2, P{w[+mC]=UAS-2xEGFP}AH2; Dr[1]/TM3, Sb[1])-
Tub[DBD] (y[1] w[*]; P{w[+mC]=Tub-GAL4DBD.D}2; P{w[+mC]=UAS-2xEGFP}AH3)

We used the following lines to report the expressions of GAL4 and LexA drivers:

13XLexAop2-IVS-myr::tdTomato-p10 in attP40,
13XLexAop2-IVS-myr::GFP in VK00005,
10XUAS-IVS-GFP-p10 in attP2 (*Pfeiffer et al., 2010*).

SS20872 line was built during this study by combining VT041832[AD] and VT029740[DBD] fly lines (*Tirian and Dickson, 2017*).

## Transgenic animals

To Convert the MIMIC lines (*Nagarkar-Jaiswal et al., 2015*) to split-GAL4 lines, we used the Trojan method (*Diao et al., 2015*). The following MIMIC lines were used:

*ChAT*-AD/DBD (MI08244): y[1] w[*]; Mi{y[+mDint2]=MIC} ChAT[MI08244] VAChT[MI08244] RRID:BDSC_55439
*VGlut*-AD:y[1] w[67c23]; Mi{PT-GFSTF.2}VGlut[MI04979-GFSTF.2] RRID:BDSC_59411
*gad1*-DBD:y[1] w[*]; Mi{y[+mDint2]=MIC}Gad1[MI09277]/TM3, Sb[1] Ser[1] RRID:BDSC_52090
ey-AD/DBD: y[1]; Mi{y[+mDint2]=MIC}ey[MI08729] RRID:BDSC_60819
*dbx*-AD/DBD: y[1] w[*]; Mi{y[+mDint2]=MIC}Dbx[MI05316] RRID:BDSC_42311
*lim3*-DBD: y[1] w[*]; Mi{y[+mDint2]=MIC}Lim3[MI03817]/SM6a RRID:BDSC_43867
*msh*-AD: [1] w[*]; Mi{y[+mDint2]=MIC}Dr[MI14348]/TM3, Sb[1] Ser[1] RRID:BDSC_59504

pBS-KS-attB2-SA()-T2A-Gal4DBD-Hsp70 or pBS-KS-attB2-SA()-T2A-p65AD-Hsp70 vectors (Addgene) with the proper reading frame were inserted to the MIMIC locus of the lines listed above via recombinase mediated cassette exchange through injection (Rainbow Transgenic Flies, Inc). Stocks were generated with the transformed flies as described in *Diao et al. (2015)*.

### Nkx6-DBD transgenic line

Nkx6-T2A-DBD was made by CRISPR/Cas technology with improved Golic+ (*Chen et al., 2015*). To knock-in the T2A-DBD coding sequence immediately preceding the *Nkx6* stop codon, 5′ and 3′ homology arms approximately 1.5 kb in length were amplified from genomic DNA with the following primers:

Nkx6_55_AgeI: ACGTACCGGTTGTCTGTGCACTCTCCCTGC;
Nkx6_53_StuI: ACGTAGGCCTATGACGATAATTATCCTGCTGCTGCT;
Nkx6_35_BamHI: ACGTGGATCCTAAAACGAATTTACAAACTATGCAATGACATGGAC;
Nkx6_33_MluI: ACGTACGCGTGAGGAGTGAAGCCTCTCTTTCGTTTAATAG,

and cloned into pTL2. T2A-DBD was then introduced by cloning DBD from pBPZpGAL4DBDUw (*Pfeiffer et al., 2010*) into pTL2 (StuI/SacI) using a forward primer containing the T2A coding sequence. gRNA target site targeting Nkx6 was finally added by annealing two primers (TCGAAA TTAAGTCTTCAGAAGA and AACTCTTCTGAAGACTTAATTT) and put into the SapI-digested vector. To target Nkx6 on chromosome 3L, this {Donor, gRNA} DNA was integrated at attP40 to create the corresponding transgenic fly, which then underwent improved Golic + for germline induced CRISPR/Cas and lethality selection for gene targeting candidates. Surviving candidates were subjected to chromosomal mapping and genomic PCR confirmation.

## RNA in situ hybridization

We employed two different techniques of RNA in situ hybridization. One technique performed smFISH (*Long et al., 2017*) to visualize mRNAs of *ChAT*, *VGlut*, and gad1in the same nervous system

(*Figure 1*). The same probes from this study were used with the following dye conjugations: DyLight550 conjugated to the *ChAT* probe, Cy5 conjugated to the *VGlut* probe, Alexa Fluor 594 conjugated to the *gad1* probe. Tissue preparation and hybridization were performed as described in *Long et al. (2017)*.

The second technique performed in situ HCR (*Choi et al., 2018*), which employs hybridization chain reaction to detect low level ChAT mRNAs in GABAergic and glutamatergic neurons (*Figure 7*). A 10-probe set targeting the entire ChAT coding region with B3-546 amplifier were purchased from www.moleculartechnologies.org. Tissues were prepared as described in *Long et al. (2017)*, but the NaBH4 treatment was skipped. We also included proteinase K digestion. Simply, after acetic acid treatment, tissues were washed with PBS for 10 min and then incubated in 10 microgram/ml Proteinase K solution for 10 min. Tissues were washed again in PBS and fixed for 55 min in 2% paraformaldehyde. After washing fixed tissues with PBS with 0.5% Tween 20, probe hybridization and amplification were performed as described in *Choi et al. (2018)*.

## Image analysis
A Zeiss LSM 710 was used to collect confocal images. z-stacks were collected with 40X lens with optical sections at 0.5 μm intervals. Image processing was performed via FIJI (http://fiji.sc/Fiji). We collected cell number data by marking each cell once via point tool in FIJI.

## Acknowledgments

We thank A Diantonio, J Skeath, C Doe, S Carroll, H Lawal, U Walldorf, H Bellen, B Dickson, G Rubin for sharing reagents. We are indebted to T Laverty, K Hibbard, A Cavallaro and the Janelia FlyCore for fly husbandry, and Alison Howard for administrative support. We thank Geoffrey Meissner and Oz Malkesman for their technical help. We thank D Shepherd for his advices in this study and J Skeath for feedback on the manuscript. This research is supported by HHMI. RHS received salary support from NIH NS083086.

## Additional information

### Competing interests
Robert H Singer: Reviewing editor, *eLife*. The other authors declare that no competing interests exist.

### Funding

| Funder | Grant reference number | Author |
| --- | --- | --- |
| Howard Hughes Medical Institute | | Tzumin Lee<br>James W Truman |
| National Institutes of Health | NS083086 | Robert H Singer |

The funders had no role in study design, data collection and interpretation, or the decision to submit the work for publication.

### Author contributions
Haluk Lacin, Conceptualization, Validation, Investigation, Visualization, Methodology, Writing—original draft, Writing—review and editing; Hui-Min Chen, Xi Long, Robert H Singer, Resources, Methodology; Tzumin Lee, Resources, Methodology, Writing—review and editing; James W Truman, Conceptualization, Supervision, Funding acquisition, Investigation, Visualization, Methodology, Writing—original draft, Writing—review and editing

### Author ORCIDs
Haluk Lacin http://orcid.org/0000-0003-2468-9618
Robert H Singer http://orcid.org/0000-0002-6725-0093
James W Truman http://orcid.org/0000-0002-9209-5435

Decision letter and Author response
Decision letter https://doi.org/10.7554/eLife.43701.017
Author response https://doi.org/10.7554/eLife.43701.018

## Additional files

### Supplementary files
• Transparent reporting form
DOI: https://doi.org/10.7554/eLife.43701.015

### Data availability
All data generated or analysed during this study are included in the manuscript and supporting files.

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
