## [Decision Letter]

Thank you for submitting your article "Neurotransmitter identity is acquired at the stem cell level in the *Drosophila* CNS" for consideration by *eLife*. Your article has been reviewed by K VijayRaghavan as the Senior Editor, a Reviewing Editor, and two reviewers. The following individuals involved in the review of your submission have agreed to reveal their identity: Sonia Sen (Reviewer #1); Matthias Landgraf (Reviewer #2).

The reviewers have discussed the reviews with one another and the Reviewing Editor has drafted this decision to help you prepare a revised submission.

Summary:

This study examines the fundamental question of how neuronal progenitor cells and the lineages that these give rise to relate to the specification of neurotransmitters. The authors investigate this question in the *Drosophila* adult ventral nervous system, focused on the fast acting small molecule transmitters, ACh, GABA, and glutamate. To this end, the authors use sophisticated genetics (previously developed by the authors and their collaborators) to visualise each post-embryonic neuroblast hemi-lineage in turn while assaying for the expression of these three fast-acting neurotransmitters (assayed by detection of GABA, ChAT and vGlut using antibodies and in situ hybridisation or by genetic reporters for GAD1, ChAT and vGlut expression). Here, the authors have undertaken to describe the neurotransmitter (NT) identity of each those postembryonic lineages. They find a simple rule governs this choice: all neurons born from a single 'hemi-lineage' share the same NT identity.

They demonstrate this through high-quality in situs to detect glutamatergic, GABAergic and cholinergic identities and show that NTs are (1) clustered, and (2) mutually exclusive in the *Drosophila* VNC. Based on the clustering they propose a possible (hemi-)lineage-based mechanism underlying NT identity; based on the mutual exclusivity, they propose that lineages must make only one type of NT neurons.

They go on to demonstrate this by relying on old and new NB-specific tools (molecular markers/Gal4s/LexAs/split lines/reporter immortalization) combined with antibody labeling for NTs. They also do the converse: use (old and new) NT-specific gal4s/lexAs/split lines to assign NT identity, and identify lineages using molecular markers, cell body position and axon tracts.

Importantly, they often use multiple approaches/reagents to demonstrate NT identity and demonstrate the exclusivity of NT identity for each NB hemi-lineage.

Finally, in the process, Lacin et al. have once again generated excellent reagents and body of reference, both of which will be invaluable for the community. The study is exceptionally well done and beautifully presented. This is an outstanding description deserving of publication in *eLife*.

Essential revisions:

Our concerns are largely to do with text and ease of digesting this wealth of information.

1) Is the Title is appropriate for the paper? This arises for two reasons: (1) The title implies a more mechanistic dissection of NT identity acquisition, and (2) if hemi-lineages have different identities, then the actual 'decision' of NT identity is more likely to be occurring in the GMC (though the NB might predispose the hemi-lineages). The authors may want to change the title accordingly.

2) The results are presented under headings 'glutamatergic lineages' etc. and under these, the authors focus on select hemi-lineages. This makes the manuscript feel a bit circular – as if the authors already know the NT identity, and they are merely confirming it. This is especially true in a couple of cases, where they use NT identity to distinguish hemi-lineages. Are the authors using their in-situs to short-list hemi-lineages present within the region of the NT clusters? Or was their search initially unbiased, and they have classified post-hoc? Either way, it will help to discuss the approach they took to start with to avoid this (clearly false) impression.

3) Can the authors explicitly tackle how they identify lineages and hemi-lineages, to begin with? They mention this only briefly in the Materials and methods section, but it would convince the reader that it's possible to unambiguously identify lineages and hemi-lineages based on cell body position, axon tracts, and molecular markers, especially if the options are restricted through the use of Gal4 lines etc. This is especially important because through the manuscript we mostly see only cell bodies (which is absolutely fine).

4) Figure 2 was extremely helpful in navigating the data, but it would be even better if the authors could add Table 1 to it – maybe a rectangle for each NB to indicate which driver was used for that NB?

5 In a few cases, the order of the text and the figures are out of sync. I think it would be much nicer if they followed each other. For example, in the 'visualizing adult lineages' section:

– The text mentions lineage-specific molecular markers – it would be nice to start with an example of that.

– The text then mentions generation of split lines based on these molecular markers – the authors show this very nicely.

– Next, the text mentions reporter immortalization – which in the current Figure is the first image in the panel. (They use this Figure to demonstrate clustering, whereas I think 1E makes that point much better). This is also true for the Figures related to the glutamatergic lineages.

---

## [Author Response]

[…] Essential revisions:Our concerns are largely to do with text and ease of digesting this wealth of information.

We appreciate the favorable evaluation of this study and suggested revisions. To address the concerns and suggestions of the reviewers, we modified the article (see below).

1) Is the Title appropriate for the paper? This arises for two reasons: (1) The title implies a more mechanistic dissection of NT identity acquisition, and (2) if hemi-lineages have different identities, then the actual 'decision' of NT identity is more likely to be occurring in the GMC (though the NB might predispose the hemi-lineages). The authors may want to change the title accordingly.

We thank the reviewers for pointing this out. The original title could imply that all neurons arising from the same stem cell use the same neurotransmitter, which is not true. We changed the Title to avoid this issue.

Original Title: “Neurotransmitter identity is acquired at the stem cell level in the *Drosophila* CNS.”

New Title: “Neurotransmitter identity is acquired in a lineage-restricted manner in the *Drosophila* CNS.”

2) The results are presented under headings 'glutamatergic lineages' etc. and under these, the authors focus on select hemi-lineages. This makes the manuscript feel a bit circular – as if the authors already know the NT identity, and they are merely confirming it. This is especially true in a couple of cases, where they use NT identity to distinguish hemi-lineages. Are the authors using their in-situs to short-list hemi-lineages present within the region of the NT clusters? Or was their search initially unbiased, and they have classified post-hoc? Either way, it will help to discuss the approach they took to start with to avoid this (clearly false) impression.

We agree with the reviewers’ concern and rewrote this part of the manuscript as shown below to indicate clearly that each hemilineage was studied without any bias.

“Utilizing the tools mentioned above we systematically visualized each hemilineage in the VNC and then mapped the fast-acting neurotransmitter used by the neurons of each hemilineage (Figure 2). We found that all neurons within a hemilineage use the same neurotransmitter (see below). Of the 34 major hemilineages per thoracic ganglia, 8 hemilineages are Glutamatergic, 12 hemilineages are GABAergic, and 14 hemilineages are cholinergic. For easier access for the readers, the data for each hemilineage will be presented below in groups based on the neurotransmitter type although the neurotransmitter identity of each hemilineage was identified independently in an unbiased fashion.”

3) Can the authors explicitly tackle how they identify lineages and hemi-lineages, to begin with? They mention this only briefly in the Materials and methods section, but it would convince the reader that it's possible to unambiguously identify lineages and hemi-lineages based on cell body position, axon tracts, and molecular markers, especially if the options are restricted through the use of Gal4 lines etc. This is especially important because through the manuscript we mostly see only cell bodies (which is absolutely fine).

We apologize for not making clear how we identified hemilineages. We identified them based on their cell body position and axonal projection. All of these features were nicely shown in three studies (Harris, 2013; Harris et al., 2015; Shepherd et al., 2016) for all the hemilineages except 9B neurons, which we introduced in this study.

We added the following sentence to indicate how hemilineages were identified.

“We employed several different approaches to visualize specific hemilineages in the adult VNC, each of which can be identified based on cell body position and axonal projection (Harris, 2013; Harris et al., 2015; Shepherd et al., 2016).”

4) Figure 2 was extremely helpful in navigating the data, but it would be even better if the authors could add Table 1 to it – maybe a rectangle for each NB to indicate which driver was used for that NB?

We thank the reviewers for this suggestion. We did this in an initial draft, but when driver information is added to Figure 2, the whole table becomes too crowded and confusing. Thus, we avoided doing this. Moreover, a better set of tools (not requiring immortalization) to mark adult VNC hemilineages has been generated by J. Truman and D. Shepherd and they will be published soon.

5 In a few cases, the order of the text and the figures are out of sync. I think it would be much nicer if they followed each other. For example, in the 'visualizing adult lineages' section:- The text mentions lineage-specific molecular markers – it would be nice to start with an example of that.- The text then mentions generation of split lines based on these molecular markers – the authors show this very nicely.- Next, the text mentions reporter immortalization – which in the current Figure is the first image in the panel. (They use this Figure to demonstrate clustering, whereas I think 1E makes that point much better). This is also true for the Figures related to the glutamatergic lineages.

We thank the reviewers for pointing out this discrepancy on the order of figure items and text. As suggested by the reviewer, we included an example of a molecular marker (Hb9 expression; now Figure 1E) to introduce lineage-specific molecular markers; then we showed the examples of split-GAL4 intersections. We also moved the immortalization example to the end of the figure as suggested. We also changed the text to indicate clustering is evident based on the RNA in situ hybridization (formerly Figure 1E, now Figure 1D). We thank the reviewers for these helpful suggestions.